# Coastal phytoplankton blooms expand and intensify in the 21st century

Yanhui Dai[1,9], Shangbo Yang[1,9], Dan Zhao[1], Chuanmin Hu[2], Wang Xu[3], Donald M. Anderson[4], Yun Li[5], Xiao-Peng Song[6], Daniel G. Boyce[7], Luke Gibson[1], Chunmiao Zheng[1,8] & Lian Feng[1✉]

Phytoplankton blooms in coastal oceans can be beneficial to coastal fisheries production and ecosystem function, but can also cause major environmental problems[1,2]—yet detailed characterizations of bloom incidence and distribution are not available worldwide. Here we map daily marine coastal algal blooms between 2003 and 2020 using global satellite observations at 1-km spatial resolution. We found that algal blooms occurred in 126 out of the 153 coastal countries examined. Globally, the spatial extent (+13.2%) and frequency (+59.2%) of blooms increased significantly (P < 0.05) over the study period, whereas blooms weakened in tropical and subtropical areas of the Northern Hemisphere. We documented the relationship between the bloom trends and ocean circulation, and identified the stimulatory effects of recent increases in sea surface temperature. Our compilation of daily mapped coastal phytoplankton blooms provides the basis for global assessments of bloom risks and benefits, and for the formulation or evaluation of management or policy actions.

Phytoplankton blooms are accumulations of microscopic algae in the surface layer of fresh and marine water bodies. Although many blooms can occur naturally, nutrients linked to anthropogenic eutrophication are expected to intensify their frequency globally[2–4]. Many algal blooms are beneficial, fixing carbon at the base of the food chain and supporting fisheries and ecosystems worldwide. However, proliferations of algae that cause harm (termed harmful algal blooms (HABs)) have become a major environmental problem worldwide[5–7]. For instance, the toxins produced by some algal species can accumulate in the food web, causing closures of fisheries as well as illness or mortality of marine species and humans[8–10]. In other cases, the decay of a dense algal bloom can deplete oxygen in bottom waters, forming anoxic 'dead zones' that can cause fish and invertebrate die-offs and ecosystem restructuring, with serious consequences for the well-being of coastal communities[1,11]. Unfortunately, algal bloom frequency and distribution are projected to increase with future climate change[12,13], with some changes causing adverse effects on aquatic ecosystems, fisheries and coastal resources.

Owing to substantial heterogeneity in space and time, algal blooms are challenging to characterize on a large scale[5,14], and thus present knowledge does not allow us to answer one of the most fundamental questions: whether algal blooms have changed in recent decades on a global basis[6,15,16]. For example, although HAB events have been compiled into the UNESCO (United Nations Educational, Scientific, and Cultural Organization) Intergovernmental Oceanographic Commission Harmful Algae Event Database (HAEDAT) globally since 1985, bloom trends are difficult to resolve, owing to inconsistent sampling efforts and the diversity of the eco-environmental or socio-economic effects[6]. Alternatively, satellite data have been used to monitor the ocean surface continuously since 1997 and have enabled bloom detection in many coastal regions[17–19]. However, the technical difficulties in dealing with complex optical features across different types of coastal waters have so far prohibited their application globally[20]. To fill this knowledge gap, we developed a method to map global coastal algal blooms and used this tool to examine satellite images between 2003 and 2020, addressing three fundamental questions: (1) where and how frequently global coastal oceans have been affected by phytoplankton blooms; (2) whether the blooms have expanded or intensified over the past two decades, both globally and regionally; and (3) the identity of the potential drivers.

## Mapping global coastal phytoplankton blooms

We generated a satellite-based dataset of phytoplankton bloom occurrence to characterize the spatial and temporal patterns of algal blooms in coastal oceans globally. The dataset was derived using global, 1-km resolution daily observations from the Moderate Resolution Imaging Spectroradiometer (MODIS) onboard NASA's Aqua satellite, and all 0.76 million images acquired by this satellite mission between 2003 and 2020 were used. We developed an automated method to detect phytoplankton blooms using MODIS images (Extended Data Fig. 1) (Methods). In this study, we define a phytoplankton bloom as the accumulation of microscopic algae at the ocean surface that exhibits satellite-detectable fluorescence signals[21]. However, whether a bloom produces toxins or is harmful to humans or the marine environment is not distinguishable from satellite data. We delineated bloom-affected areas (that is, the areas where algal blooms were detected), and enumerated the bloom count

[1]School of Environmental Science and Engineering, Southern University of Science and Technology, Shenzhen, China. [2]College of Marine Science, University of South Florida, St. Petersburg, FL, USA. [3]Shenzhen Ecological and Environmental Monitoring Center of Guangdong Province, Shenzhen, China. [4]Woods Hole Oceanographic Institution, Woods Hole, MA, USA. [5]School of Marine Science and Policy, College of Earth, Ocean, and Environment, University of Delaware, Lewes, DE, USA. [6]Department of Geographical Sciences, University of Maryland, College Park, MD, USA. [7]Bedford Institute of Oceanography, Fisheries and Oceans Canada, Dartmouth, Nova Scotia, Canada. [8]EIT Institute for Advanced Study, Ningbo, China. [9]These authors contributed equally: Yanhui Dai, Shangbo Yang. ✉e-mail: fengl@sustech.edu.cn

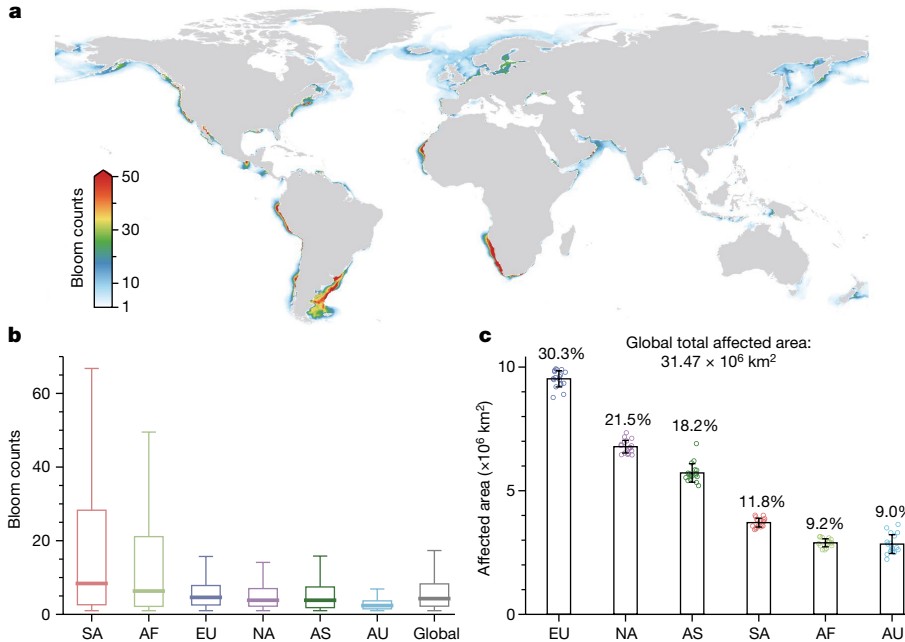

**Fig. 1 | Global patterns of coastal phytoplankton blooms between 2003 and 2020. a**, The spatial distribution of annual mean bloom count based on daily satellite detections. **b**, Continental and global statistics for annual mean bloom count (South America (SA), $n = 3,846,125$; Africa (AF), $n = 2,516,225$; Europe (EU), $n = 17,703,949$; North America (NA), $n = 10,034,286$; Asia (AS), $n = 5,371,158$; Australia (AU), $n = 2,781,998$ pixel observations). The centre line represents the median value, bottom and top bounds of boxes are first and third quartiles, and the whiskers show a maximum of 1.5 times the interquartile range. **c**, Continental statistics for the long-term annual mean of bloom-affected areas ($n = 18$ years). The percentages show the corresponding contributions to the global total. The bars represent s.d. Open circles are the affected areas during different years. Map created using Python 3.8.

at the 1-km pixel level (that is, the number of detected blooms per pixel) (Fig. 1). We further estimated the bloom frequency (dimensionless) by integrating the bloom count and affected areas within 1° × 1° grid cells (see Methods), and this metric was used to examine temporal dynamics in bloom intensity. Validation with independent satellite samples selected via several visual inspection techniques showed an overall accuracy level of more than 95% for our method, and comparisons using discrete events in HAEDAT[6] indicated that we successfully identified bloom counts for 79.3% of the historical HAB events in that database (Extended Data Figs. 2–6). We examined phytoplankton blooms in the exclusive economic zones (EEZs) of 153 coastal countries and in 54 large marine ecosystems (LMEs) (Extended Data Fig. 7). Our study area encompasses global continental shelves and outer margins of coastal currents, which offer the majority of marine resources available for human use (see Methods). Out of the 153 coastal countries examined, 126 were observed to have phytoplankton blooms (Fig. 1). The total bloom-affected area was 31.47 million km², equivalent to approximately 24.2% of the global land area and 8.6% of the global ocean area, with a median bloom count of 4.3 per year during the past 2 decades (Fig. 1b). Europe (9.52 million km²−30.3% of the total affected area) and North America (6.78 million km²−21.5% of the total affected area) contributed the largest bloom areas. By contrast, the most frequent blooms were found around Africa and South America (median bloom counts of more than 6.3 per year). Australia experienced the lowest frequency (2.4 per year) and affected area (2.84 million km²−9.0% of the total affected area) of blooms.

Phytoplankton blooms occurred frequently in the eastern boundary current systems (that is, California, Benguela, Humboldt and Canary), northeastern USA, Latin America, the Baltic Sea, Northern Black Sea and the Arabian Sea (Fig. 1a). Five LMEs were found with the most frequent blooms (annual median bloom count over 15), including Patagonian Shelf, Northeast US continental Shelf, the Baltic Sea, Gulf of California and Benguela Current (Extended Data Fig. 7). These hotspots are often reported as having a high incidence of algal blooms, some of which are HABs, driven by either coastal upwelling or pronounced anthropogenic nutrient enrichment[9,22–26]. European LMEs showed mostly large proportions of bloom-affected areas, and some also showed frequent bloom occurrences. By contrast, Asian LMEs exhibited mainly infrequent blooms, given their large affected areas. We identified more bloom events in estuarine regions than along coasts in regions without major river discharge ($P < 0.05$; Extended Data Fig. 8), highlighting the critical role of terrestrial nutrient sources in coastal algal blooms[3].

## Long-term trends

The total global bloom-affected area has expanded by 3.97 million km² (13.2%) between 2003 and 2020, equivalent to 0.14 million km² yr⁻¹ ($P < 0.05$; Fig. 2). Furthermore, the number of countries with significant bloom expansion was about 1.6 times those with a decreasing trend. The global median bloom frequency showed an increasing rate of 59.2% (+2.19% yr⁻¹, $P < 0.05$) over the observed period. Spatially, areas showing significant increasing trends ($P < 0.05$) in bloom frequency were 77.6% larger than those with the opposite trends (Fig. 2). Globally, our analysis revealed overall consistent fluctuations between the bloom-affected area and bloom frequency between 2003 and 2020 (Fig. 2b). However, there was no significant relationship between bloom extent and frequency in 23 countries and 10 LMEs over the past two decades, underscoring the spatial and temporal variability of algal blooms and the importance of continuous satellite monitoring.

The entire Southern Hemisphere was primarily characterized by increased bloom frequency, although weakened blooms were also sometimes found. In the Northern Hemisphere, the low latitude (<30° N) coasts were mainly featured with strong bloom weakening (Fig. 2a), primarily in the California Current System and the Arabian Sea. Bloom strengthening was found in the northern Gulf of Mexico and the East and South China Seas, albeit at smaller magnitudes. At higher latitudes, weakening blooms were detected mainly in the northeastern North Atlantic and the Okhotsk Sea in the northwestern North Pacific. Globally, the largest increases in bloom frequency were observed in six

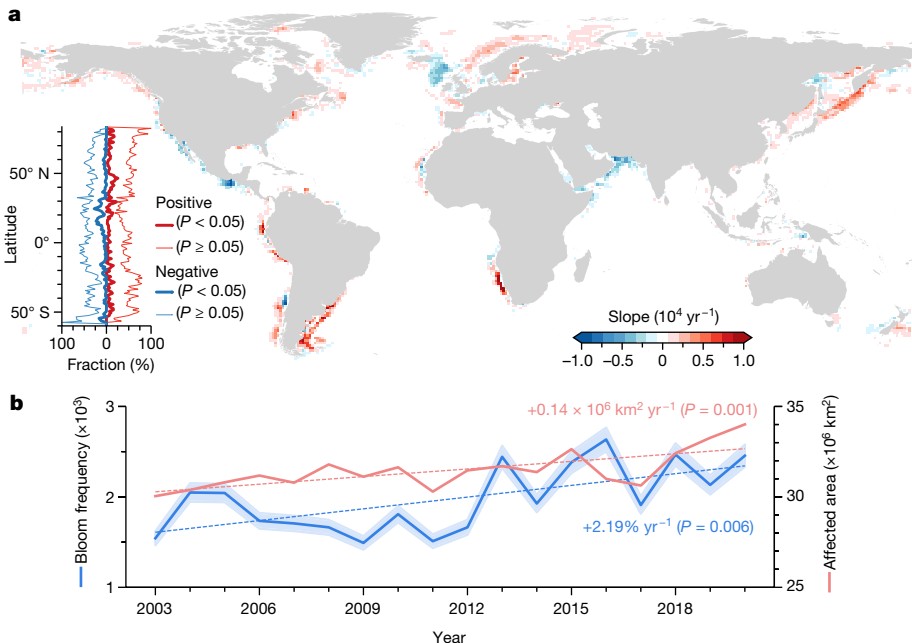

**Fig. 2 | Trends of global coastal phytoplankton blooms between 2003 and 2020. a**, Spatial patterns of the trends in bloom frequency at a 1° × 1° grid scale. The latitudinal profiles show the fractions of grids with significant and insignificant trends (positive or negative) along the east–west direction. **b**, Interannual variability and trends in annual median bloom frequency and total global bloom-affected area. The linear slopes and $P$-value (two-sided $t$-test) are indicated. The shading associated with the bloom frequency data represents an uncertainty level of 5% in bloom detection. Map created using Python 3.8.

major coastal current systems, including Oyashio (+6.31% yr⁻¹), Alaska (+5.22% yr⁻¹), Canary (+4.28% yr⁻¹), Malvinas (+3.02% yr⁻¹), Gulf Stream (+2.42% yr⁻¹) and Benguela (+2.30% yr⁻¹) (Figs. 2a and 3).

## Natural and anthropogenic effects

Increases in sea surface temperature (SST) can stimulate bloom occurrence. We found significant positive correlations ($P < 0.05$) between the annual mean bloom frequency and the coincident SST (SST data were averaged over the growth window of algal blooms within a year (Methods and Extended Data Fig. 9)) in several high-latitude regions (>40° N), such as the Alaska Current ($r = 0.44$), the Oyashio Current ($r = 0.48$) and the Baltic Sea ($r = 0.41$) (Fig. 3). These findings agree with previous studies, in which the bloom-favourable seasons in these temperate seas have been extended under warmer temperatures[27–29]. However, this temperature-based mechanism did not apply to regions with inconsistent trends between SST and bloom frequency, particularly for the substantial bloom weakening in the tropical and subtropical areas (Figs. 2a and 3b).

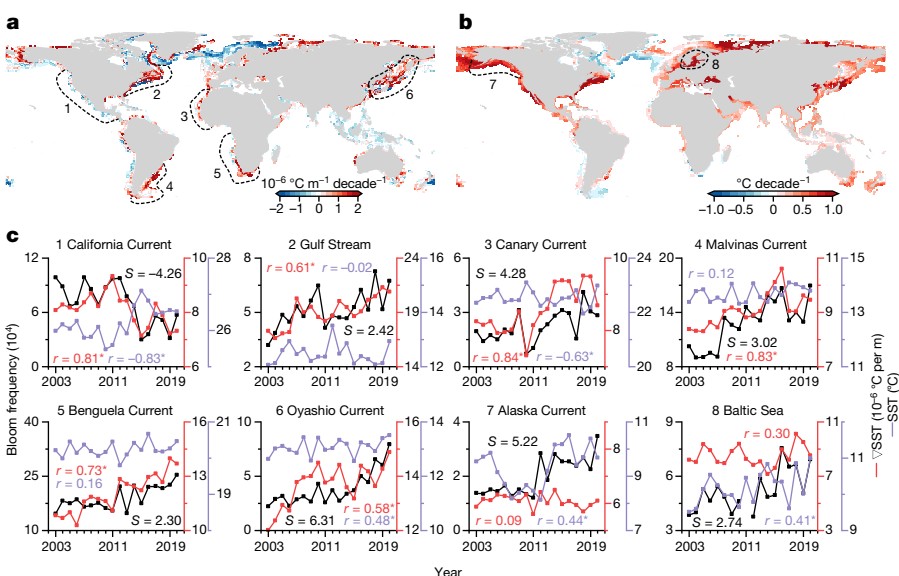

**Fig. 3 | Effects of climate change on phytoplankton blooms. a,b**, Global patterns of trends in SST gradient (**a**) and SST (**b**) from 2003 to 2020. **c**, Long-term changes in bloom frequency in the regions labelled in **a** and **b**, and their relationship to the SST and SST gradient. Linear slope ($S$) of bloom frequency and the correlation coefficient ($r$) between bloom frequency and the SST and the SST gradient (∇SST) are shown. Asterisks indicate statistically significant ($P < 0.05$) correlations. Maps created using ArcMap 10.4 and Python 3.8.

Changes in climate can also affect ocean circulation, altering ocean mixing and the transport of nutrients that drive the growth of marine phytoplankton and bloom formation[30–32]. We used the spatial SST gradient (in °C m$^{-1}$) as a proxy for the magnitude of oceanic mesoscale currents (the time-varying velocity of kinetic energy (also known as the eddy kinetic energy (EKE))) by following the methods of a previous study[33], and examined its effects on algal blooms (Methods). The trend in the SST gradient appeared more spatially aligned to bloom frequency than SST. We found significant positive correlations ($P < 0.05$) between the SST gradient and bloom frequency for various coastal current systems, including the Canary ($r = 0.84$), Malvinas ($r = 0.83$), California ($r = 0.81$), Benguela ($r = 0.73$), Gulf Stream ($r = 0.61$) and Oyashio ($r = 0.58$) currents.

Trends in bloom frequency in subtropical eastern boundary upwelling regions (the California, Benguela and Canary currents) followed the changes in mesoscale currents (Fig. 3a,c). In the California Current System, the decrease in bloom frequency was probably due to the weakened upwelling (represented by a reduced SST gradient and increased SST) and thus lower nutrient supply[25]. Conversely, the Canary and Benguela currents were characterized by strengthened upwelling and increased bloom frequency. The two western boundary current systems at high latitudes (Malvinas and Oyashio)—although characterized by less pronounced upwelling[34]—exhibited a similar mechanism to the subtropical eastern boundary regions. For the subtropical western boundary Gulf Stream current, the strengthened current jets (manifested as a larger SST gradient) brought more nutrients from the continental shelf[35], triggering more algal blooms. Nevertheless, whether these changes in oceanic mesoscale activities were responses to wind, stratification, the shear of ocean currents or other factors[33] requires region-based investigations.

Global climate events, represented as the multivariate El Niño–Southern Oscillation index[36] (MEI), also showed connections with coastal bloom frequency. The minimum MEI in 2010 (a strong La Niña year) was followed by a low bloom frequency in the following year, and the largest MEI in 2015 (a strong El Niño year) was followed by the strongest bloom frequency in 2016 (Fig. 2b and Extended Data Fig. 10a).

Changes in anthropogenic nutrient enrichment may have also contributed to the trends in blooms[37]. For example, the decline in bloom frequency in the Arabian Sea, without clear links to SST or SST gradient changes, could result from decreased fertilizer use in the surrounding countries (such as Iran) (Extended Data Fig. 10). By contrast, the bloom strengthening in some Asian countries could be attributed to surges in fertilizer use[38]. We examined trends in fertilizer usage (either nitrogen or phosphorus) and bloom frequency and found high positive correlations in China, Iran, Vietnam and the Philippines. Paradoxically, decreased fertilizer uses and increased bloom frequency were identified in some countries, suggesting that nutrient control efforts might have been counterbalanced by the stimulatory effects of climate warming or other factors. Furthermore, the intensified aquaculture industry in Finland, China, Algeria, Guinea, Vietnam, Argentina, Russia and Uruguay may also be linked to their increased bloom incidence, as suggested by the significant positive correlations ($r > 0.5$, $P < 0.05$) between their aquaculture production and bloom frequency. A similar relationship between aquaculture expansion and positive trends in HAB incidence was reported from an analysis of HAEDAT data[6]. However, analogous positive feedbacks for fertilizer or aquaculture were not found in many other countries. Thus, an ecosystem model incorporating terrestrial and oceanic nutrient transport and nutrient–plankton relationships of different species[39] is required to quantify the contributions of natural and anthropogenic factors to algal blooms[14].

## Future implications

We acknowledge that our criteria for a detectable bloom event is operationally defined by sensor sensitivities and other factors, and that the bloom count metric used here may underestimate algal bloom incidence, particularly compared to harmful events entered in HAEDAT. For example, in a recent global analysis of the HAEDAT events, Hallegraeff et al.[6] report a dozen or more events per year for each of nine regions over a 33-year study period, compared to the global median bloom count of 4.3 in this study. There are several possible explanations for this discrepancy, such as the many low-cell-concentration HABs that are not detectable from space but that can still cause harm, as well as sensor sensitivities and algorithm thresholds. Furthermore, our bloom count was averaged over all 1-km pixels within the EEZs, whereas HAEDAT entries are based on discrete sampling regions. This underestimation does not, however, alter the trends and other conclusions of this study, as the metrics used here were constant across time and space. Underestimates would have been uniform across regions globally. In this regard, it is of note that the study of Hallegraeff et al.[6] found a significant link between the number of HAEDAT events over time and the global expansion of aquaculture production, similar to findings in our study.

The major contribution of our study is to provide a spatially and temporally consistent characterization of global coastal algal blooms between 2003 and 2020. Globally, increasing trends in algal bloom area and frequency are apparent. Regionally, however, trends were non-uniform owing to the compounded effects of changes in climate (such as changes in SST and SST gradient and climate extremes), anthropogenic eutrophication and aquaculture development. Our daily mapping of bloom events offers valuable baseline information to understand the mechanisms underlying the formation, maintenance, and dissipation of algal blooms[5,40]. This could aid in developing forecasting models (on either global or regional scales) that can help minimize the consequences of harmful blooms, and can also help in policy decisions relating to the control of nutrient discharges and other HAB-stimulatory factors. Noting again that many blooms are beneficial, particularly in terms of their positive effects on ecosystems as well as on wild and farmed fisheries, the results here can also contribute toward policies and management actions that sustain those beneficial blooms.

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

## Methods

### Data sources

MODIS on the Aqua satellite provides a global coverage within 1–2 days. All images acquired by this satellite mission from January 2003 to December 2020 were used in our study to detect global coastal phytoplankton blooms, with a total of 0.76 million images. MODIS Level-1A images were downloaded from the Ocean Biology Distributed Active Archive Center (OB.DAAC) at NASA Goddard Space Flight Center (GSFC), and were subsequently processed with SeaDAS software (version 7.5) to obtain Rayleigh-corrected reflectance ($R_{rc}$ (dimensionless), which was converted using the rhos (in $sr^{-1}$) product (rhos × π) from SeaDAS)[41], remote sensing reflectance ($R_{rs}$ ($sr^{-1}$)) and quality control flags (l2_flags). If a pixel was flagged by any of the following, it was then removed from phytoplankton bloom detection: straylight, cloud, land, high sunglint, high solar zenith angle and high sensor zenith angle (https://oceancolor.gsfc.nasa.gov/atbd/ocl2flags/). MODIS level-3 product for aerosol optical thicknesses (AOT) at 869 nm was also obtained from OB.DAAC NASA GSFC (version R2018.0), which was used to examine the impacts of aerosols on bloom trends.

We examined the algal blooms in the EEZs of 153 ocean-bordering countries (excluding the EEZs in the Caspian Sea or around the Antarctic), 126 of which were found with at least one bloom in the past two decades. The EEZ dataset is available at https://www.marineregions.org/download_file.php?name=World_EEZ_v11_20191118.zip. The EEZs are up to 200 nautical miles (or 370 km) away from coastlines, which include all continental shelf areas and offer the majority of marine resources available for human use. Regional statistics of algal blooms were also performed for LMEs. LMEs encompass global coastal oceans and outer edges of coastal currents areas, which are defined by various distinct features of the oceans, including hydrology, productivity, bathymetry and trophically dependent populations[42]. Of the 66 LMEs identified globally, we excluded the Arctic and Antarctic regions and examined 54 LMEs. The boundaries of LMEs were obtained from https://www.sciencebase.gov/catalog/item/55c77722e4b08400b1fd8244.

We used HAEDAT to validate our satellite-detected phytoplankton blooms in terms of presence or absence. The HAEDAT dataset (http://haedat.iode.org) is a collection of records of HAB events, maintained under the UNESCO Intergovernmental Oceanographic Commission and with data archives since 1985. For each HAB event, the HAEDAT records its bloom period (ranging from days to months) and geolocation. We merged duplicate entries when both the recorded locations and times of the HAEDAT events were very similar to one another, and a total number of 2,609 HAEDAT events were ultimately selected between 2003 and 2020.

We used the ¼° resolution National Oceanic and Atmospheric Administration Optimum Interpolated SST (v. 2.1) data to examine the potential simulating effects of warming on the global phytoplankton trends. We also estimated the SST gradients following the method of Martínez-Moreno[33]. As detailed in ref. [33], the SST gradient can be used as a proxy for the magnitude of oceanic mesoscale currents (EKE). We used the SST gradient to explore the effects of ocean circulation dynamics on algal blooms.

Fertilizer uses and aquaculture production for different countries was used to examine the potential effects of nutrient enrichment from humans on global phytoplankton bloom trends. Annual data between 2003 and 2019 on synthetic fertilizer use, including nitrogen and phosphorus, are available from https://ourworldindata.org/fertilizers. Annual aquaculture production includes cultivated fish and crustaceans in marine and inland waters, and sea tanks, and the data between 2003 and 2018 are available from https://ourworldindata.org/grapher/aquaculture-farmed-fish-production.

The MEI, which combines various oceanic and atmospheric variables[36], was used to examine the connections between El Niño–Southern Oscillation activities and marine phytoplankton blooms. The dataset is available from https://psl.noaa.gov/enso/mei/.

### Development of an automated bloom detection method

A recent study by the UNESCO Intergovernmental Oceanographic Commission revealed that globally reported HAB events have increased[6]. However, such an overall increasing trend was found to be highly correlated with recently intensified sampling efforts[6]. Once this potential bias was accounted for by examining the ratio between HAB events to the number of samplings[5], there was no significant global trend in HAB incidence, though there were increases in certain regions. With synoptic, frequent, and large-scale observations, satellite remote sensing has been extensively used to monitor algal blooms in oceanic environments[17–19]. For example, chlorophyll *a* (Chl*a*) concentrations, a proxy for phytoplankton biomass, has been provided as a standard product by NASA since the proof-of-concept Coastal Zone Color Scanner (1978–1986) era[43,44]. The current default algorithm used to retrieve Chl*a* products is based on the high absorption of Chl*a* at the blue band[45,46], which often shows high accuracy in the clear open oceans but high uncertainties in coastal waters. This is because, in productive and dynamic coastal oceans, the absorption of Chl*a* in the blue band can be obscured by the presence of suspended sediments and/or coloured dissolved organic matter (CDOM)[47]. To address this problem, various regionalized Chl*a* algorithms have been developed[48]. Unfortunately, the concentrations of the water constituents (CDOM, sediment and Chl*a*) can vary substantially across different coastal oceans. As a result, a universal Chl*a* algorithm that can accurately estimate Chl*a* concentrations in global coastal oceans is not currently available.

Alternatively, many spectral indices have been developed to identify phytoplankton blooms instead of quantifying their bloom biomass, including the normalized fluorescence line height[21] (nFLH), red tide index[49] (RI), algal bloom index[47] (ABI), red–blue difference (RBD)[50], *Karenia brevis* bloom index[50] (KBBI) and red tide detection index[51] (RDI). In practice, the most important task for these index-based algorithms is to determine their optimal thresholds for bloom classification. However, such optimal thresholds can be regional-or image-specific[20], due to the complexity of optical features in coastal waters and/or the contamination of unfavourable observational conditions (such as thick aerosols, thin clouds, and so on), making it difficult to apply spectral-index-based algorithms at a global scale.

To circumvent the difficulty in determining unified thresholds for various spectral indices across global coastal oceans, an approach from a recent study to classify algal blooms in freshwater lakes[52] was adopted and modified here. In that study, the remotely sensed reflectance data in three visible bands (red, green and blue) were converted into two-dimensional colour space created by the Commission Internationale de l'éclairage (CIE), in which the position on the CIE chromaticity diagram represented the colour perceived by human eyes (Extended Data Fig. 1a). As the algal blooms in freshwater lakes were manifested as greenish colours, the reflectance of bloom-containing pixels was expected to be distributed in the green gamut of the CIE chromaticity diagram; the stronger the bloom, the closer the distance to the upper border of the diagram (the greener the water).

Here, the colour of phytoplankton blooms in the coastal oceans can be greenish, yellowish, brownish, or even reddish[53], owing to the compositions of bloom species (diatoms or dinoflagellates) and the concentrations of different water constituents. Furthermore, the Chl*a* concentrations of the coastal blooms are typically lower than those in inland waters, thus demanding more accurate classification algorithms. Thus, the algorithm proposed by Hou et al.[52] was modified when using the CIE chromaticity space for bloom detection in marine environments. Specifically, we used the following coordinate conversion formulas to obtain the *xy* coordinate values in the CIE colour space:

$$x = X/(X+Y+Z)$$
$$y = Y/(X+Y+Z)$$
$$X = 2.7689R + 1.7517G + 1.1302B \quad (1)$$
$$Y = 1.0000R + 4.5907G + 0.0601B$$
$$Z = 0.0000R + 0.0565G + 5.5943B$$

where $R$, $G$ and $B$ represent the $R_{rc}$ at 748 nm, 678 nm (fluorescence band) and 667 nm in the MODIS Aqua data, respectively. By contrast, the $R$, $G$ and $B$ channels used in Hou et al.[52] were the red, green and blue bands. We used the fluorescence band for the $G$ channel because, for a given region, the 678 nm signal increases monotonically with the Chl$a$ concentration for blooms of moderate intensity[21], which is similar to the response of greenness to freshwater algal blooms. Thus, the converted $y$ value in the CIE coordinate system represents the strength of the fluorescence. In practice, for pixels with phytoplankton blooms, the converted colours in the chromaticity diagram will be located within the green, yellow or orange–red gamut (see Extended Data Fig. 1a); the stronger the fluorescence signal is, the closer the distance to the upper border of the CIE diagram (larger $y$ value). By contrast, for bloom-free pixels without a fluorescence signal, their converted $xy$ coordinates will be located in the blue or purple gamut. Therefore, we can determine a lower boundary in the CIE two-dimensional coordinate system to separate bloom and non-bloom pixels, similar to the method proposed by Hou et al.[52].

We selected 53,820 bloom-containing pixels from the MODIS $R_{rc}$ data as training samples to determine the boundary of the CIE colour space. These sample points were selected from nearshore waters worldwide where frequent phytoplankton blooms have been reported (Extended Data Fig. 2); the algal species included various species of dinoflagellates and diatoms[20]. A total of 80 images was used, which were acquired from different seasons and across various bloom magnitudes, to ensure that the samples used could almost exhaustively represent the different bloom conditions in the coastal oceans.

We combined the MODIS FLH$_{Rrc}$ (fluorescence line height based on $R_{rc}$) and enhanced red–green–blue composite (ERGB) to delineate bloom pixels manually. The FLH$_{Rrc}$ image was calculated as:

$$FLH_{Rrc} = R_{rc678} \times F_{678} - [R_{rc667} \times F_{667} + (R_{rc748} \times F_{748} - R_{rc667} \times F_{667}) \times (678 - 667)/(748 - 667)] \quad (2)$$

where $R_{rc667}$, $R_{rc678}$ and $R_{rc748}$ are the $R_{rc}$ at 667, 678 and 748 nm, respectively, and $F_{667}$, $F_{678}$ and $F_{748}$ are the corresponding extraterrestrial solar irradiance. ERGB composite images were generated using $R_{rc}$ of three bands at 555 (R), 488 (G) and 443 nm (B). Although phytoplankton-rich and sediment-rich waters have high FLH$_{Rrc}$ values, they appear as darkish and bright features in the ERGB images (Extended Data Fig. 3), respectively[21]. In fact, visual examination with fluorescence signals and ERGB has been widely accepted as a practical way to delineate coastal algal blooms on a limited number of images[21,54,55]. Note that the FLH$_{Rrc}$ here was slightly different from the NASA standard nFLH product[56], as the latter is generated using $R_{rs}$ (corrected for both Rayleigh and aerosol scattering) instead of $R_{rc}$ (with residual effects of aerosols). However, when using the NASA standard algorithm to further perform aerosol scattering correction over $R_{rc}$, 20.7% of our selected bloom-containing pixels failed to obtain valid $R_{rs}$ (without retrievals or flagged as low quality), especially for those with strong blooms (see examples in Extended Data Fig. 4). Likewise, we also found various nearshore regions with invalid $R_{rs}$ retrievals. By contrast, $R_{rc}$ had valid data for all selected samples and showed more coverage in nearshore coastal waters. The differences between $R_{rs}$ and $R_{rc}$ were because the assumptions for the standard atmospheric correction algorithm do not hold for bloom pixels or nearshore waters with complex optical properties[57]. In fact, $R_{rc}$ has been used as an alternative to $R_{rs}$ in various applications in complex waters[58,59].

We converted the $R_{rc}$ data of 53,820 selected sample pixels into the $xy$ coordinates in the CIE colour space (Extended Data Fig. 1a). As expected, these samples of bloom-containing pixels were located in the upper half of the chromaticity diagram (the green, yellow and orange–red gamut) (Extended Data Fig. 1a). We determined the lower boundary of these sample points in the chromaticity diagram, which represents the lightest colour and thus the weakest phytoplankton blooms; any point that falls above this boundary represents stronger blooms. The method to determine the boundary was similar to Hou et al.[52]: we first binned the sample points according to the $x$ value in the chromaticity diagram and estimated the 1st percentile ($Q_1$%) of the corresponding $Y$ for each bin; then, we fit the $Q_1$% using two-order polynomial regression. Sensitivity analysis with $Q_{0.3}$% (the three-sigma value) resulted in minor changes (<1%) in the resulting bloom areas for single images. Notably, sample points were rarely located near white points ($x = 1/3$ and $y = 1/3$, represent equal reflection from three RGB bands) in the diagram, and we used two polynomial regressions to determine the boundaries for $x$ values greater and less than 1/3, which can be expressed as:

$$y_1 = 4.8093x^2 - 3.0958x + 0.8357 \quad x < \frac{1}{3} \quad (3)$$

$$y_2 = 4.9040x^2 - 3.5759x + 0.9862 \quad x > \frac{1}{3} \quad (4)$$

Based on the above, if a pixel's $xy$ coordinate (converted from $R_{rc}$ spectrum) satisfies the conditions of ($x < 1/3$ AND $y > y_1$) or ($x > 1/3$ AND $y > y_2$), it is classified as a 'bloom' pixel.

Depending on the local region and application purpose, the meaning of 'phytoplankton bloom' may differ. Here, for a global application, the pixelwise bloom classification is based on the relationship (represented using the CIE colour space) between $R_{rc}$ in the 667-, 678- and 754-nm bands derived from visual interpretation of the 80 pairs of FLH$_{Rrc}$ and ERGB imagery. Instead of a simple threshold, we used a lower boundary of the sample points in the chromaticity diagram to define a bloom. In simple words, a pixel is classified as a bloom if its fluorescence signal is detectable (the associated $xy$ coordinate in the CIE colour space located above the lower boundary). Histogram of the nFLH values from the 53,820 training pixels demonstrated the minimum value of -0.02 mW cm$^{-2}$ μm$^{-1}$ (Extended Data Fig. 1a), which is in line with the lower-bound signal of $K. brevis$ blooms on the West Florida shelf[21,47]. Note that, such a minimum nFLH is determined from the global training pixels, and it does not necessarily represent a unified lower bound for phytoplankton blooms across the entire globe, especially considering that fluorescence efficiency may be a large variable across different regions. Different regions may have different lower bounds of nFLH to define a bloom, and such variability is represented by the predefined boundary in the CIE chromaticity diagram in our study. Correspondingly, although the accuracy of Chl$a$ retrievals may have large uncertainties in coastal waters, the histogram of the 53,820 training pixels shows a lower bound of -1 mg m$^{-3}$ (Extended Data Fig. 1a). Similarly to nFLH, such a lower bound may not be applicable to all coastal regions, as different regions may have different lower bounds of Chl$a$ for bloom definition.

Although the MODIS cloud (generated by SeaDAS with $R_{rc869} < 0.027$) and associated straylight flags can be used to exclude most clouds, we found that residual errors from thin clouds or cloud shadows could affect the spectral shape and cause misclassification for bloom detections. Thus, we designed two spectral indices to remove such effects:

$$Index1 = nR_{rc488} - nR_{rc443} - (nR_{rc555} - nR_{rc443}) \times 0.5 \quad (5)$$

$$Index2 = nR_{rc555} - nR_{rc469} - (nR_{rc645} - nR_{rc469}) \times 0.5 \quad (6)$$

where Index1 and Index2 were used to remove cloud shadows and clouds, respectively. The $nR_{rc443}$, $nR_{rc488}$ and $nR_{rc555}$ in index1 are the normalized $R_{rc}$, obtained by normalizing $R_{rc488}$. Similar calculations were performed for index2. The purpose of normalizations is to eliminate the effect of the absolute magnitude of the reflectance, so that the thresholds of these two indices are influenced by only the relative magnitude (spectral shape). We determined thresholds for Index1 (>0.12) and Index2 (<0.012) through trial-and-error and ensured that the misclassifications caused by residual errors from clouds and cloud shadows could be effectively removed. After applying the cloud/cloud shadow and various other masks that are associated with l2_flags, we obtained an annual mean valid pixel observation ($N_{vobs}$) of ~2.0 × 10⁵ for global 1° × 1° grid cells, and the fluctuation patterns and trends of $N_{vobs}$, either annually or seasonally, are different from that of the global bloom frequency and affected areas (see Supplementary Fig. 1).

## Assessments of the algorithm performance

In addition to phytoplankton blooms, macroalgal blooms (*Sargassum* and *Ulva*) frequently occur in many coastal oceans[60–63]. To verify whether our CIE-fluorescence algorithm could eliminate such impacts, we compared the spectra between micro-and macroalgal blooms (see Extended Data Fig. 1b). We found that the spectral shapes of *Sargassum* and *Ulva* are substantially different from those of microalgae, particularly for the three bands used for CIE coordinate conversion. The converted *xy* coordinates for macroalgae were located in the purple–red gamut of the CIE diagram, which was far below the predefined boundary (Extended Data Fig. 1). Moreover, our algorithm is not affected by highly turbid waters for the following two reasons: first, extremely high turbidity tends to saturate the MODIS ocean bands[64], which can be easily excluded; second, without a fluorescence peak, the reflectance of unsaturated turbid waters, after conversion to CIE coordinates, will be located below the predefined boundary (see example in Extended Data Fig. 1b). We also confirmed that the spectral shapes of coccolithophore blooms are different from dinoflagellates and diatoms (see example in Extended Data Fig. 1b), and thus they are excluded from our algorithm.

Three different types of validation methods were adopted to demonstrate the reliability of the proposed CIE-fluorescence algorithm for phytoplankton bloom detection in global coastal oceans, including visual inspections of the RGB, ERGB and $FLH_{Rrc}$ images, verifications using independent manually delineated algal blooms, and comparisons with the reported HAB events from the HAEDAT dataset.

First, we selected MODIS Aqua images from different locations where coastal phytoplankton blooms have been recorded in the published literature. We visually compared the RGB, ERGB, and $FLH_{Rrc}$ images, and our algorithm detected bloom patterns (see examples in Extended Data Fig. 3). Comparisons from various images worldwide showed that our algorithm could successfully identify regions with high $FLH_{Rrc}$ values and brownish-to-darkish features on the ERGB images, which can be considered phytoplankton blooms.

Second, we delineated additional 15,466 bloom-containing pixels from 35 images covering different coastal areas, using the same visual inspection and manual delineation method as for the training sample pixels. Moreover, we also selected 14,149 bloom-free pixels (bright or turquoise green colours on ERGB images or low $FLH_{Rrc}$ values) within the same images as bloom-containing images. We applied our algorithm to all these pixels, and compared the algorithm-identified and manually delineated results. Our CIE-fluorescence algorithm showed high values in both producer and user accuracies (92.04% and 98.63%) (Supplementary Table 1), and appeared effective at identifying bloom pixels and excluding false negatives (blooms classified as non-blooms) and false positives (non-blooms classified as blooms).

Third, we validated the satellite-detected phytoplankton blooms using in situ reported HAB events from the HAEDAT dataset. For each HAB event in the HAEDAT dataset, we obtained all MODIS images over the reported bloom period (from days to months). Within each year, we estimated the ratio between the number of satellite images with 'bloom detected' against the number of valid images (see definition above) during the bloom periods across the entire globe (Supplementary Table 1). Moreover, we calculated the number of events with at least one successful satellite bloom detection ($N_s$), and then estimated the ratio between $N_s$ and the total HAB events for each year. Results showed that substantial amounts (averaged at 51.2%) of satellite observations acquired during the HAB event periods were found with phytoplankton blooms by our algorithm. Overall, 79.3% of the global HAB events were successfully identified by satellite. The discrepancies between satellite and in situ observations could be explained by the following reasons: first, our study focused only on the phytoplankton blooms that are resolvable by satellite fluorescence signals; other types of HAB events in the HAEDAT dataset may not have been detectable by satellite observations, such as events with lower phytoplankton biomass but high toxicity, occurrences at the subsurface layers, or fluorescence signals overwhelmed by suspended sediments[65–67]. Second, although the HAEDAT recorded HAB events could be sustained for long periods, high biomass of surface algae may not have occurred every day within this period due to the changes in stratification, precipitation, wind, vertical migration of cells, and many other factors[68]. Third, the spatial scale of certain HAB events may have been too small to be identified using the 1-km resolution MODIS observations. Fourth, a reduced MODIS satellite observation frequency by the contaminations of clouds and land adjacency effects[69]. Therefore, we believe the underestimations of satellite-detected blooms compared to the in situ reported HAB events were mainly due to inconsistencies between the two observations rather than uncertainties in our algorithm.

Because $R_{rc}$ depends not only on water colour but also on aerosols (type and concentration) and solar and viewing geometry, a sensitivity analysis was used to determine whether such variables could impact bloom detection. Aerosol reflectance ($\rho_a$) with different AOTs at 869 nm was simulated using the NASA-recommended maritime aerosol model (r75f02, with a relative humidity of 75% and a fine-mode fraction of 2%). Then, $\rho_a$ of each MODIS band was added to $R_{rc}$ images, and the resulting bloom areas with and without added $\rho_a$ were compared. Results showed that even with a change of 0.02 in AOT at 869 nm, the bloom areas showed minor changes (<2%) in the tested images; minor changes were also found when we used different aerosol models to conduct $\rho_a$ simulations[70]. Note that 0.02 represents the high end of the AOT intra-annual variability in coastal oceans (see Extended Data Fig. 5), and the associated interannual changes are much smaller. Thus, the use of $R_{rc}$ instead of the fully atmospherically corrected reflectance $R_{rs}$ could have limited impacts on our detected global bloom trend.

We also tried various index-based algorithms developed in previous studies. However, results showed that all these methods require image-specific thresholds to accurately determine algal bloom boundaries for different coastal regions (see Extended Data Fig. 6). By contrast, although our CIE-fluorescence algorithm may lead to different bloom thresholds for different regions, it can identify bloom pixels without adjusting the coefficients and, therefore, is more suitable for global-scale bloom assessment efforts.

We acknowledge that our satellite-detected algal blooms represent only high amounts of phytoplankton biomass on the ocean surfaces without distinguishing whether such blooms produce toxins or are harmful to marine environments. Furthermore, with only limited spectral information from MODIS, it is difficult to discriminate the phytoplankton species of algal blooms; such information could help to improve our understanding of the impacts of these phytoplankton blooms. However, we expect these challenges to be addressed soon with the scheduled launch of the Plankton, Aerosol, Cloud, ocean Ecosystem (PACE) mission by NASA in 2024, where the hyperspectral measurements over a broad spectrum (350–885 nm) will make species-level classifications possible[71].

## Exploring the patterns and trends of global coastal phytoplankton blooms

We applied the CIE-fluorescence algorithm to all MODIS Aqua level-2 $R_{rc}$ images, and a total number of 0.76 million images between 2003 and 2020 were processed. We mapped all detected blooms into 1-km daily scale level-3 composites. The number of bloom counts within a year for each location can be easily enumerated, and the long-term annual mean values were then estimated (Fig. 1a). We further calculated the total global bloom-affected area (the areas where algal blooms were detected at least once) for each year and examined their changes over time (Fig. 2b).

We defined bloom frequency (dimensionless) to represent the density of phytoplankton blooms for a year by integrating the bloom count and bloom-affected areas within 1°×1° grid cells within that year, which is expressed as:

$$\text{Bloom frequency} = \frac{n}{N} \sum_{i=1}^{n} M_i \tag{7}$$

where $M_i$ is the enumerated bloom count for each 1-km resolution pixel in a year within one 1° × 1° grid cell, and $n$ represents the associated number of bloom-affected pixels in the same cell (the number of pixels with $M_i > 0$), and $N$ is the total number of 1-km MODIS pixels in this grid cell. We estimated the bloom frequency for each year between 2003 and 2020, and determined the long-term trend over global EEZs through a linear least-squares regression (see Fig. 2a).

Continental and country-level statistics were performed for bloom count, bloom-affected areas, and bloom frequency (Fig. 1b,c and Supplementary Table 2), using boundaries for the EEZs of different ocean-bordering countries (see above). Similar statistics were also conducted for 54 LMEs (Extended Data Fig. 7 and Supplementary Table 3).

## Correlations with SST and SST gradient

To assess the impacts of climate change on long-term trends in coastal phytoplankton blooms, we correlated the annual mean bloom frequency and the associated SST and SST gradient in various coastal current systems for grid cells with significant changes in bloom frequency (Fig. 3c). The SST and SST gradient were averaged over the growth window within a year, assuming that the changes within the growth window, either in water temperatures or ocean circulations, play more important roles in the bloom trends compared to other seasons[32].

We determined the growth window of phytoplankton blooms for each 1° × 1° grid cell (Extended Data Fig. 9a) using the following method: first, we estimated the proportion of cumulative bloom-affected pixels within the grid cells for a year. Second, a generalized additive model[72] was used to determine the shape of the phenological curves (Extended Data Fig. 9b), where a log link function and a cubic cyclic regression spline smoother were applied[73,74]. Third, the timing of maximum bloom-affected areas (TMBAA) was then determined by identifying the inflection point on the bloom growth curve (Extended Data Fig. 9c). To facilitate comparisons across Northern and Southern Hemispheres, the year in the Southern Hemisphere was shifted forward by 183 days (Extended Data Fig. 9c). We characterized the similarity of the bloom growth curve between different grid cells and grouped them into three distinct clusters using a fuzzy c-means cluster analysis method[75,76]. We found uniform distributions of the clusters over large geographic areas. Cluster I is mainly distributed in mid-low latitudes (<45° N and <30° S), where the maximum bloom-affected areas were expected in the early period of the year. Cluster II was mostly found in higher latitudes, with bloom developments (quasi-) synchronized with increases in SST. Cluster III was detected along the coastlines, where the bloom-affected areas increase throughout the entire year. In practice, the growth window for clusters I and III was set as the entire year, and that for cluster II

was set from day 150 to day 270 within the year. We further found that the TMBAA for cluster II showed small changes over the entire period (Extended Data Fig. 9d), indicating relatively stable phenological cycles for those phytoplankton blooms[32,77].

## Reporting summary

Further information on research design is available in the Nature Portfolio Reporting Summary linked to this article.

## Data availability

The satellite-based dataset of global coastal algal bloom at 1-km resolution and the associated code are available at https://doi.org/10.5281/zenodo.7359262. Source data are provided with this paper.

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

**Acknowledgements** We thank NASA for providing global MODIS satellite images, and the Intergovernmental Oceanographic Commission (IOC) of UNESCO for providing the HAEDAT dataset. L.F. was supported by the National Natural Science Foundation of China (no. 41890852, 42271322 and 41971304). D.M.A. was supported by the Woods Hole Center for Oceans and Human Health (National Science Foundation grant OCE-1840381 and National Institutes of Health grants NIEHS-1P01-ES028938-01).

**Author contributions** Y.D. and S.Y.: methodology, data processing and analyses, and writing. L.F.: conceptualization, methodology, funding acquisition, supervision and writing. D.Z.: data processing and analysis. C.H., W.X., D.M.A., Y.L., X.-P.S., D.G.B., L.G. and C.Z. participated in interpreting the results and refining the manuscript.

**Competing interests** The authors declare no competing interests.

**Additional information**
**Correspondence and requests for materials** should be addressed to Lian Feng.

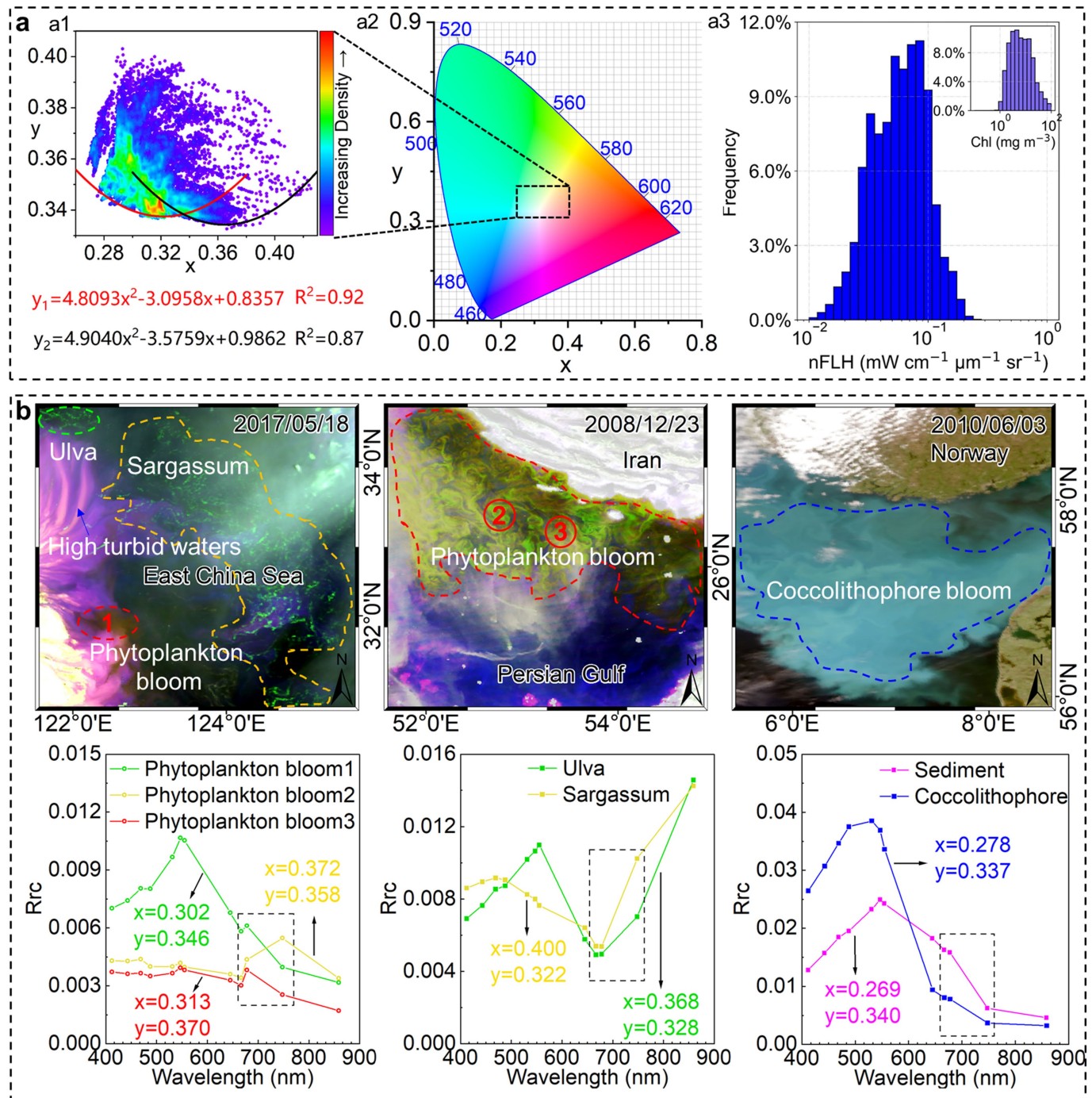

**Extended Data Fig. 1 | Development of the CIE-fluorescence algorithm to detect phytoplankton blooms using MODIS satellite imagery. (a)**. A1: The density plot of manually delineated bloom-containing pixels in the CIE coordinate system ($n$ = 53,820), and their distribution in the CIE color space (box in A2). A3: Histograms of nFLH and Chla for the delineated pixels, obtained using NASA standard algorithms[47,57]. (**b**) MODIS true color composites and selected spectra for phytoplankton blooms, macroalgal blooms (*Ulva* and *Sargassum*), coccolithophore blooms, and sediment-rich turbid waters. The x-y numbers indicate their corresponding positions in the CIE coordinate system. The black rectangular boxes in the three lower panels highlight different spectral shapes between phytoplankton blooms and other features near the fluorescence band. Maps created using ArcMap 10.4.

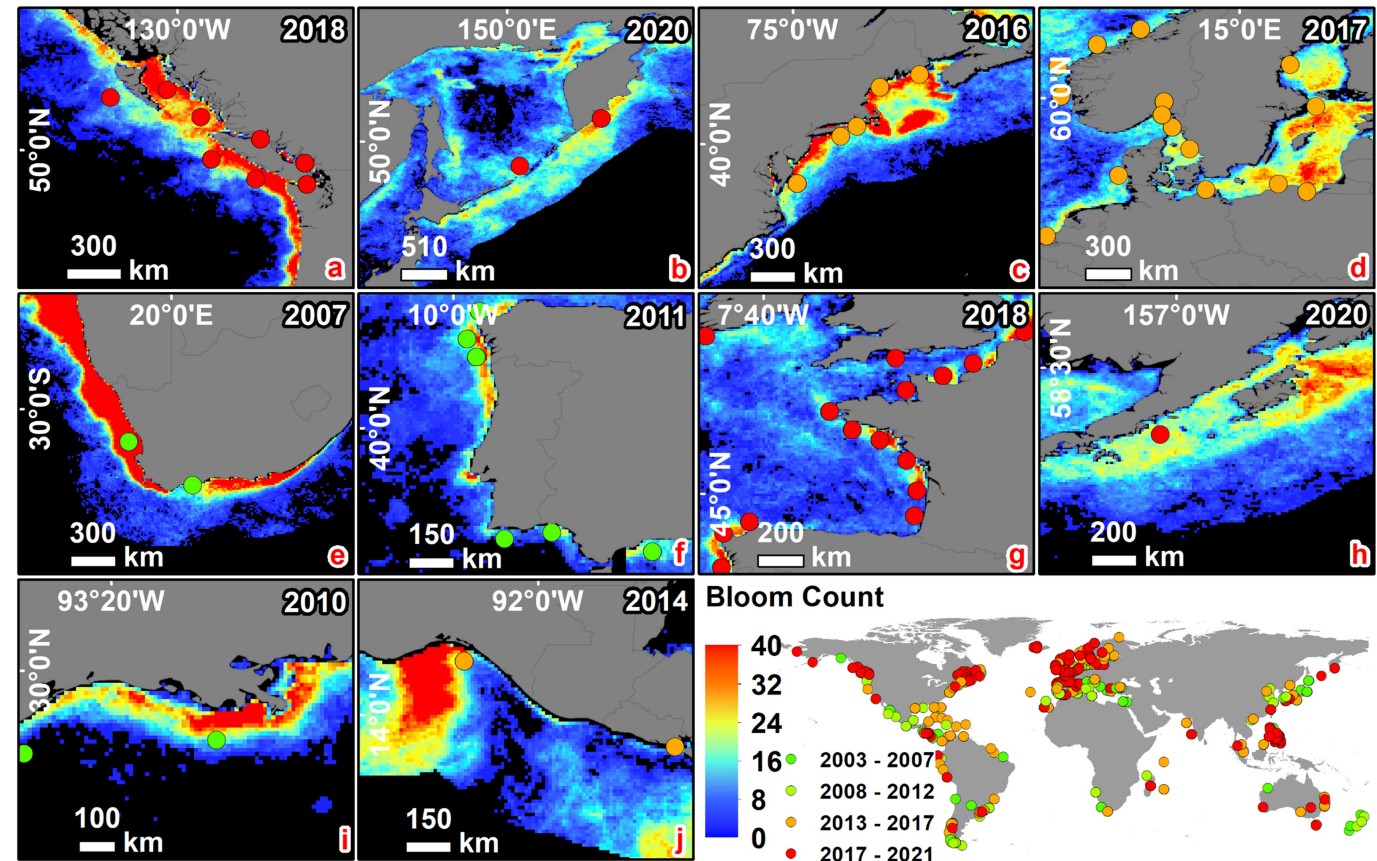

**Extended Data Fig. 2 | MODIS-detected bloom count within certain years for several coastal regions with frequently reported blooms.** The MODIS observational year is annotated within each panel, and overlaid points indicate *in situ* recorded harmful algal bloom events from the Harmful Algae Event Database (HAEDAT) within the same year. The lower right panel shows the locations of all the HAEDAT records that were used for algorithm validations in this study (Supplementary Table 1), which also demonstrates the increase in sampling effort in the most recent years. Created using ArcMap 10.4.

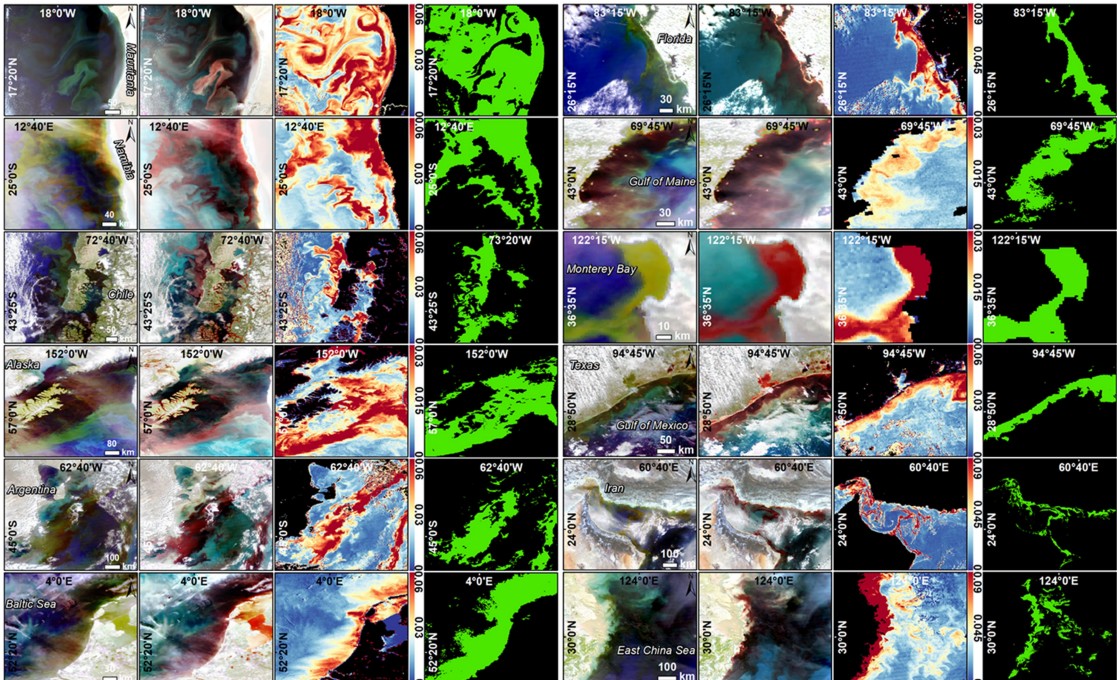

**Extended Data Fig. 3 | Performance of the CIE-fluorescence algorithm for phytoplankton bloom detection in 12 selected coastal oceans.** From left to right are the RGB-true color composite, ERGB composite, FLHRrc, and the bloom area (green pixels) detected by the CIE-fluorescence algorithm. Created using ArcMap 10.4.

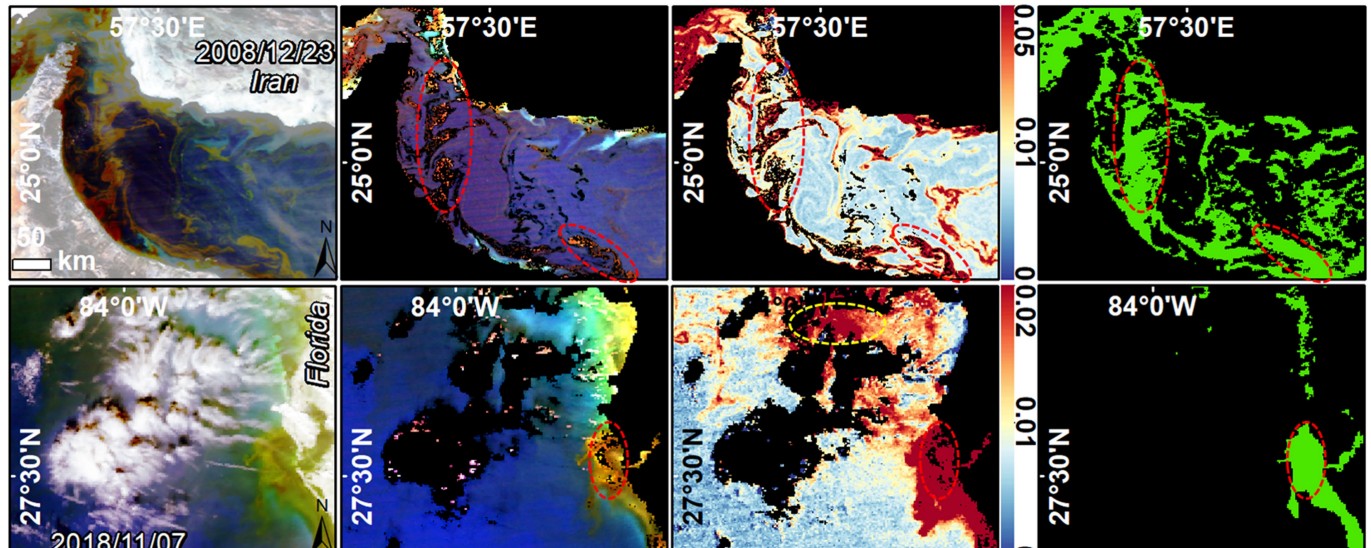

**Extended Data Fig. 4 | Examples showing disadvantages of using NASA standard $R_{rs}$ (i.e., with the removal of both Rayleigh and aerosol scattering) in algal bloom detection.** From left to right are the RGB composites, ERGB, nFLH, and the bloom areas (green pixels) detected by the CIE-fluorescence algorithm (based on $R_{rc}$, without the removal of aerosol scattering). Substantial amounts of invalid $R_{rs}$ retrievals can be observed in the red-encircled areas in which severe blooms can be found. Additionally, nFLH shows high values at cloud edges (yellow-encircled areas), making it challenging to use a simple threshold to classify blooms. However, such problems can be circumvented in our CIE-fluorescence algorithm. Created using ArcMap 10.4.

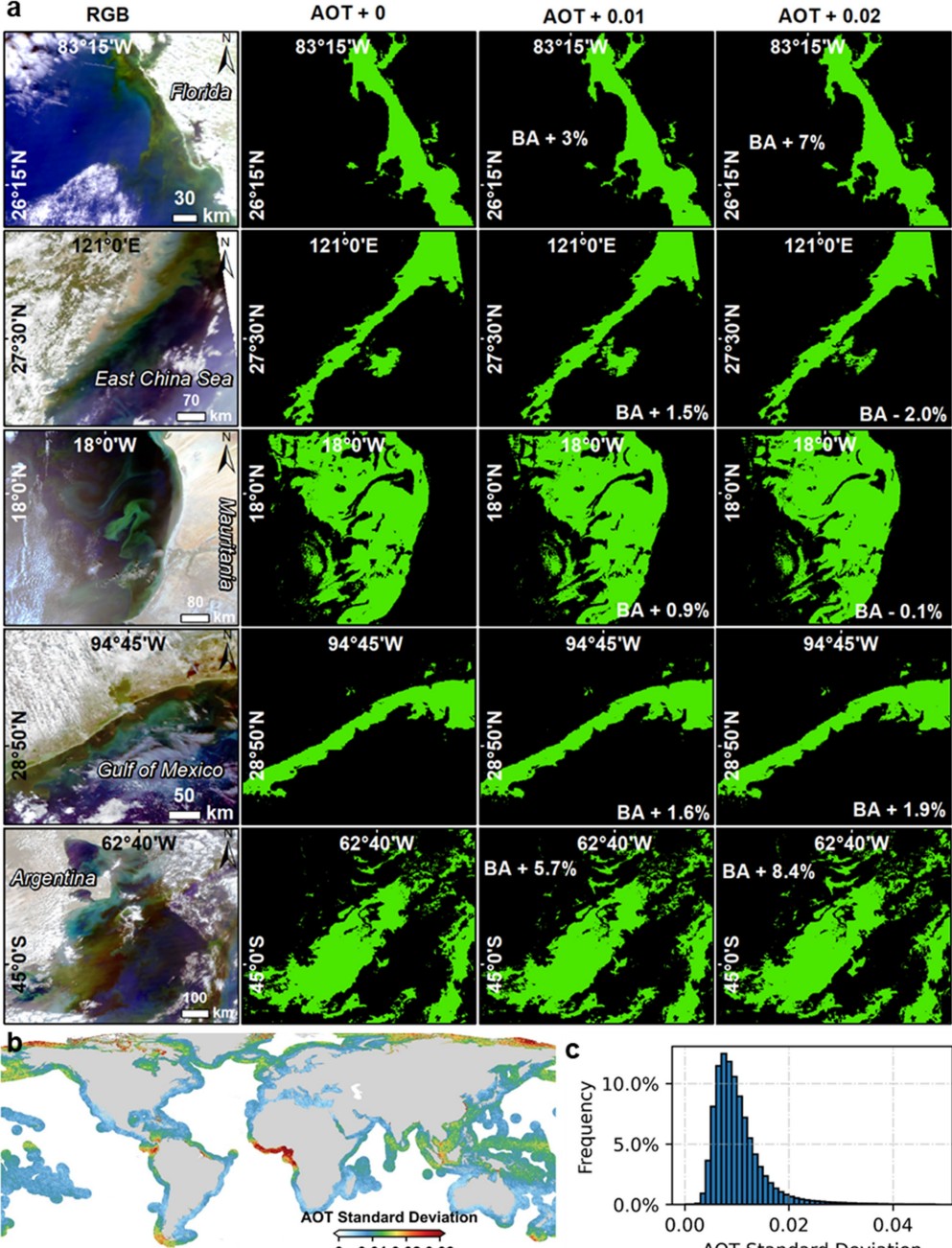

**Extended Data Fig. 5 | Sensitivity analysis of the impacts of aerosols on bloom detection.** (**a**) Responses of bloom area (BA) to changes in aerosol optical thickness (AOT). Aerosol reflectance ($\rho_a$) with AOTs of 0.01 and 0.02 at 869-nm is simulated and added to the MODIS images, and the resulting bloom areas (green pixels) with and without added $\rho_a$ are compared. The left columns show the RGB composites, and the right three columns show the bloom areas under different AOTs. The percentages of BA changes are annotated in the panels. (**b**) The standard deviation between the 12 monthly mean values of AOT in global coastal waters (i.e., 66.7% of the intra-annual variability), and the histogram is shown in (**c**). Maps created using ArcMap 10.4.

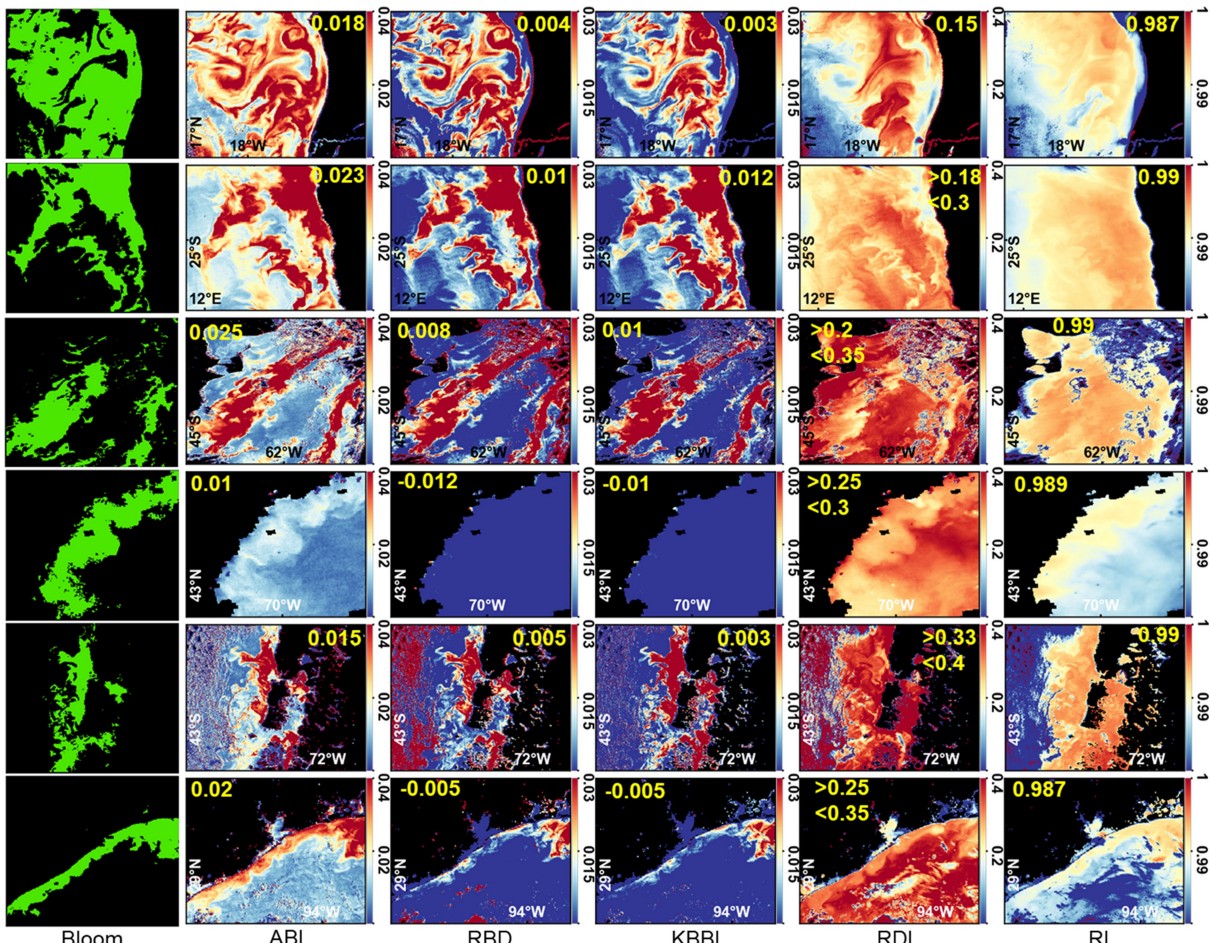

| Bloom | ABI | RBD | KBBI | RDI | RI |
|---|---|---|---|---|---|

**Extended Data Fig. 6 | Comparison of different index-based algorithms in algal bloom detection in various coastal regions.** Image-specific thresholds (annotated within the panels) are required (labeled within the panels) for RI[50], ABI (estimated with FLH$_{Rrc}$)[48], RBD[51], KBBI[51], and RDI[52] to delineate accurate bloom areas (i.e., high nFLH values, which appear as bright and darkish features on the ERGB images). The left panels are the bloom areas (green pixels) extracted using our CIE-fluorescence algorithm. The RGB-true color and ERGB composites are shown in Extended Data Fig. 3. Created using ArcMap 10.4.

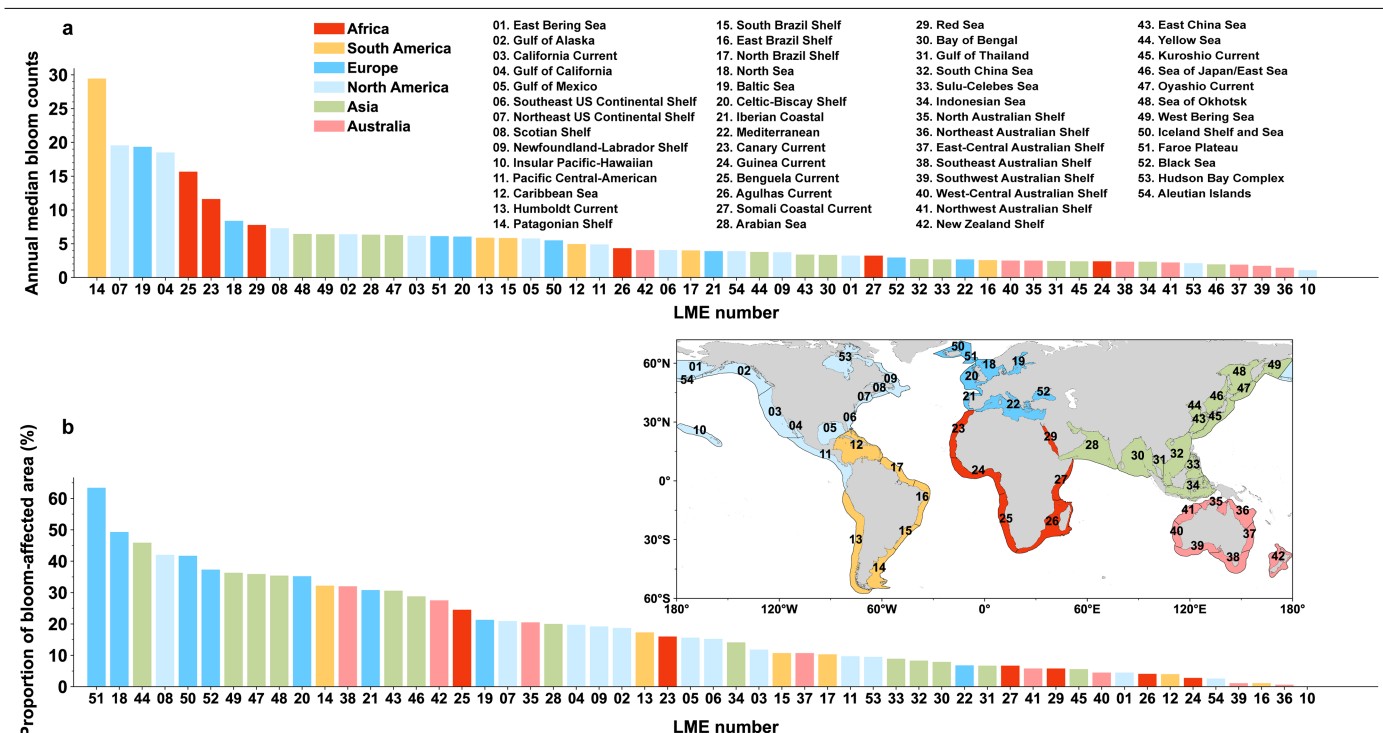

**Extended Data Fig. 7 | Annual median bloom count and the proportion of bloom-affected areas for large marine ecosystems (LMEs).** (**a**) Annual median bloom count, (**b**) proportion of bloom-affected areas. The data are ordered from the largest to the smallest. The LMEs are grouped by continent, and their names, numbers, and locations are shown in (a) and (b). Map created using Python 3.8.

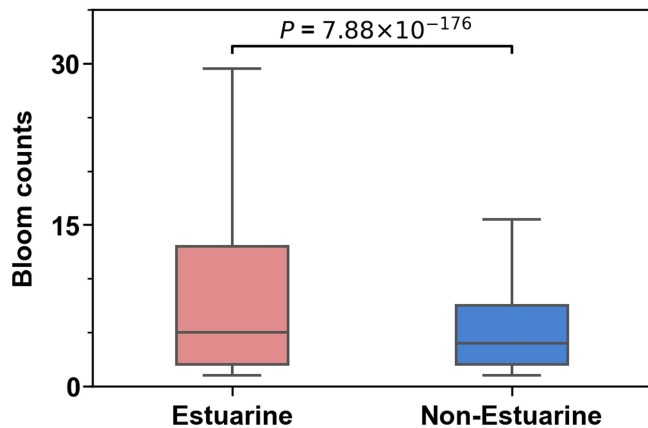

$P = 7.88 \times 10^{-176}$

**Extended Data Fig. 8 | Comparison of bloom counts in the estuarine and non-estuarine regions.** Boxplots for long-term mean bloom count in the estuarine (n = 13,622 pixel observations) and non-estuarine (n = 361,604 pixel observations) regions. Comparison analysis was performed by two sided Welch's t-test ($P < 0.001$). Upper and lower bounds are first and third quartiles, the bar in the middle represents the median value, and the whiskers show the minimum and maximum values. Sixty-two estuarine zones from large rivers were selected, and the boundary of each zone was manually delineated according to high-resolution satellite images.

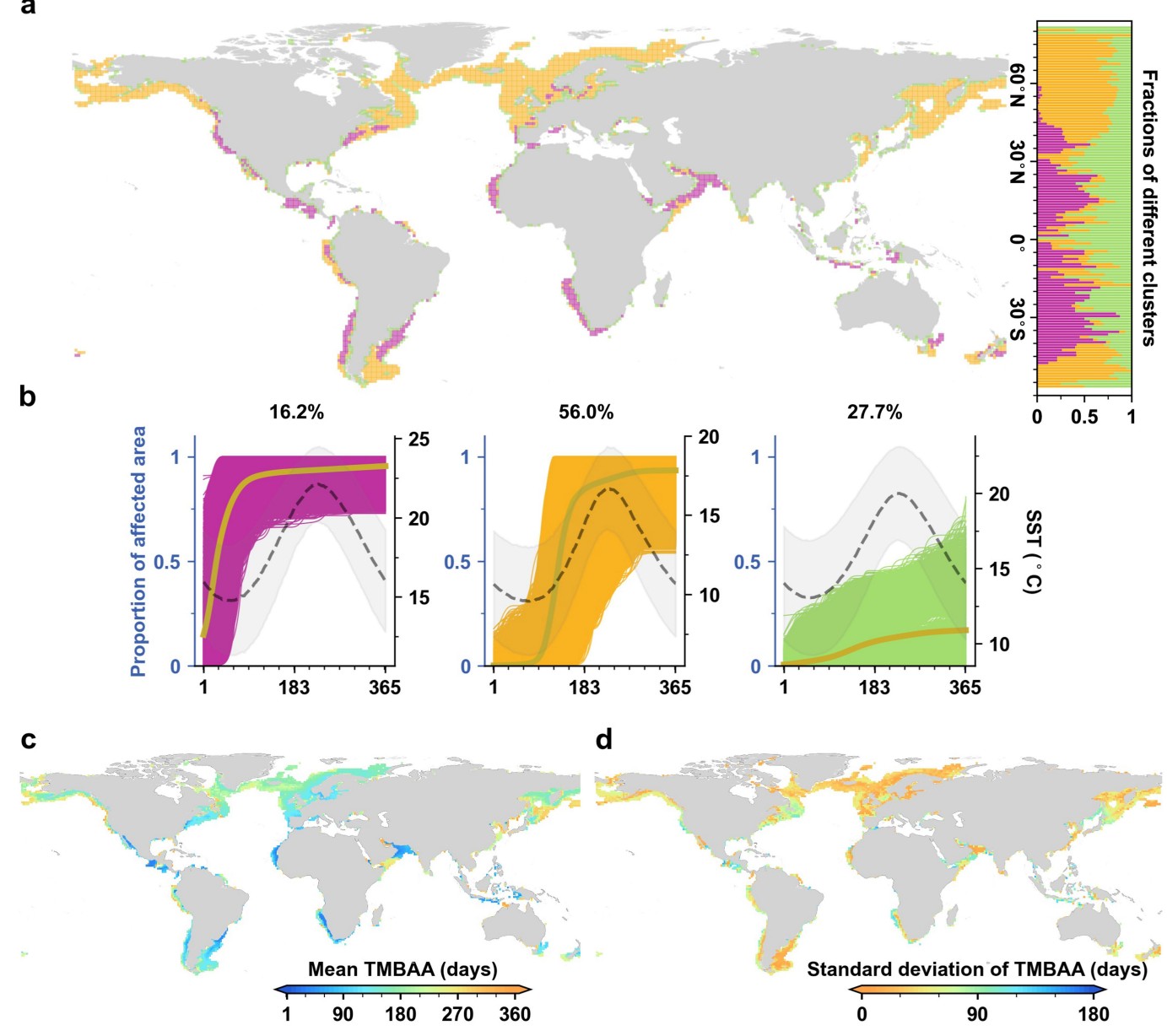

**Extended Data Fig. 9 | Clusters of different bloom growth paths. (a)** The spatial distribution of different clusters. The fractions of different clusters across different latitudes are summarized. (**b**) The development of the maximum bloom-affected areas within a year within 1° × 1° grid cells, where all global grid cells are grouped into three distinct clusters according to the similarity of the bloom growth curve. The colored bond curves represent the mean values of all the grid cells, and their mean SST and associated standard deviations are shown with dashed lines and gray shading. The proportions of different clusters in the global bloom-affected areas are annotated. (**c**) and (**f**) The mean timing of the maximum bloom-affected areas (TMBAA) and the associated standard deviations between 2003 and 2019. The whole year in the Southern Hemisphere is shifted forward by 183 days in (**c**). Maps created using Python 3.8.

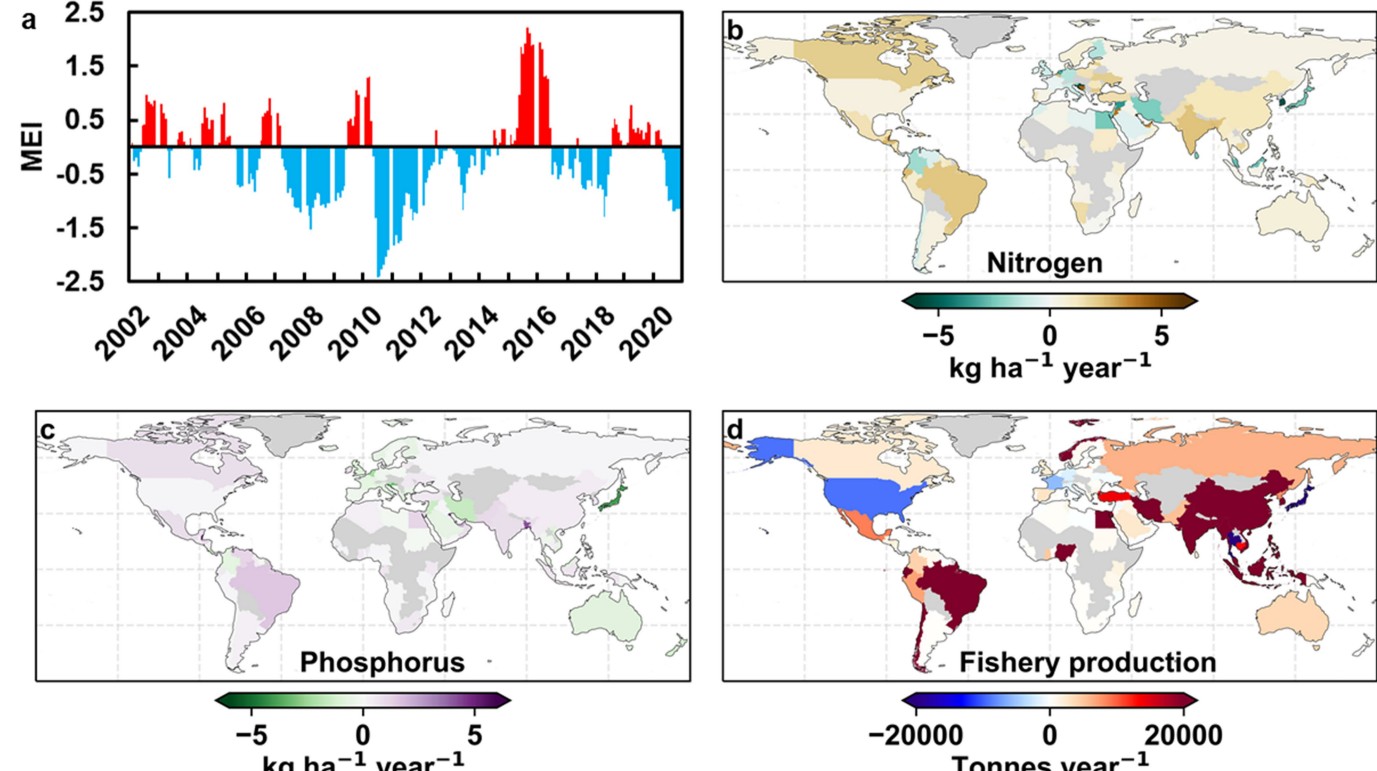

**Extended Data Fig. 10 | Changes in climate extremes, global fertilizer uses, and fishery production over the past two decades. (a)** Changes in the bi-monthly Multivariate El Niño–Southern Oscillation (ENSO) index (MEI) between 2002 and 2020. Positive and negative MEI values represent El Niño and La Niña events, respectively. The dots show annual mean values. **(b–c)** Trends of nitrogen and phosphorus from 2003 to 2019 for different countries. **(d)** Trends of fishery production from 2003 to 2018. Gray indicates no data. Maps created using ArcMap 10.4.

# Reporting Summary

## Statistics

For all statistical analyses, confirm that the following items are present in the figure legend, table legend, main text, or Methods section.

| n/a | Confirmed | |
|---|---|---|
| ☐ | ☒ | The exact sample size (*n*) for each experimental group/condition, given as a discrete number and unit of measurement |
| ☐ | ☒ | A statement on whether measurements were taken from distinct samples or whether the same sample was measured repeatedly |
| ☐ | ☒ | The statistical test(s) used AND whether they are one- or two-sided *Only common tests should be described solely by name; describe more complex techniques in the Methods section.* |
| ☒ | ☐ | A description of all covariates tested |
| ☒ | ☐ | A description of any assumptions or corrections, such as tests of normality and adjustment for multiple comparisons |
| ☐ | ☒ | A full description of the statistical parameters including central tendency (e.g. means) or other basic estimates (e.g. regression coefficient) AND variation (e.g. standard deviation) or associated estimates of uncertainty (e.g. confidence intervals) |
| ☐ | ☒ | For null hypothesis testing, the test statistic (e.g. *F*, *t*, *r*) with confidence intervals, effect sizes, degrees of freedom and *P* value noted *Give P values as exact values whenever suitable.* |
| ☒ | ☐ | For Bayesian analysis, information on the choice of priors and Markov chain Monte Carlo settings |
| ☐ | ☒ | For hierarchical and complex designs, identification of the appropriate level for tests and full reporting of outcomes |
| ☒ | ☐ | Estimates of effect sizes (e.g. Cohen's *d*, Pearson's *r*), indicating how they were calculated |

*Our web collection on statistics for biologists contains articles on many of the points above.*

## Software and code

Policy information about availability of computer code

| Data collection | The satellite data were obtained from the U.S. National Aeronautics and Space Administration (NASA) Goddard Space Flight Center (GSFC). |
|---|---|
| Data analysis | SeaDAS (Version 7.5) were used to analyze the satellite images. |

For manuscripts utilizing custom algorithms or software that are central to the research but not yet described in published literature, software must be made available to editors and reviewers. We strongly encourage code deposition in a community repository (e.g. GitHub). See the Nature Portfolio guidelines for submitting code & software for further information.

## Data

Policy information about availability of data

All manuscripts must include a data availability statement. This statement should provide the following information, where applicable:
- Accession codes, unique identifiers, or web links for publicly available datasets
- A description of any restrictions on data availability
- For clinical datasets or third party data, please ensure that the statement adheres to our policy

The MODIS Aqua data can be obtained from the U.S. National Aeronautics and Space Administration (NASA) Goddard Space Flight Center (GSFC).
The in situ reported HAB data are available from events from http://haedat.iode.org.
The Exclusive economic zones (EEZs) dataset is available at https://www.marineregions.org/download_file.php?name=World_EEZ_v11_20191118.zip.
The boundaries of large marine ecosystems (LMEs) were obtained from https://www.sciencebase.gov/catalog/item/55c77722e4b08400b1fd8244.

Annual data between 2003 and 2019 on synthetic fertilizer use, including nitrogen and phosphorus, are available from https://ourworldindata.org/fertilizers.
Annual aquaculture production includes cultivated fish and crustaceans in marine and inland waters, and sea tanks, and the data between 2003 and 2018 are available from https://ourworldindata.org/grapher/aquaculture-farmed-fish-production.
The dataset is available from https://psl.noaa.gov/enso/mei/.

## Human research participants

Policy information about studies involving human research participants and Sex and Gender in Research.

| | |
|---|---|
| Reporting on sex and gender | *Use the terms sex (biological attribute) and gender (shaped by social and cultural circumstances) carefully in order to avoid confusing both terms. Indicate if findings apply to only one sex or gender; describe whether sex and gender were considered in study design whether sex and/or gender was determined based on self-reporting or assigned and methods used. Provide in the source data disaggregated sex and gender data where this information has been collected, and consent has been obtained for sharing of individual-level data; provide overall numbers in this Reporting Summary. Please state if this information has not been collected. Report sex- and gender-based analyses where performed, justify reasons for lack of sex- and gender-based analysis.* |
| Population characteristics | *Describe the covariate-relevant population characteristics of the human research participants (e.g. age, genotypic information, past and current diagnosis and treatment categories). If you filled out the behavioural & social sciences study design questions and have nothing to add here, write "See above."* |
| Recruitment | *Describe how participants were recruited. Outline any potential self-selection bias or other biases that may be present and how these are likely to impact results.* |
| Ethics oversight | *Identify the organization(s) that approved the study protocol.* |

Note that full information on the approval of the study protocol must also be provided in the manuscript.

# Field-specific reporting

Please select the one below that is the best fit for your research. If you are not sure, read the appropriate sections before making your selection.

☐ Life sciences ☐ Behavioural & social sciences ☒ Ecological, evolutionary & environmental sciences

For a reference copy of the document with all sections, see nature.com/documents/nr-reporting-summary-flat.pdf

# Ecological, evolutionary & environmental sciences study design

All studies must disclose on these points even when the disclosure is negative.

| | |
|---|---|
| Study description | This study developed a novel method to map global coastal algal blooms and used this tool to examine satellite images between 2003 and 2020, addressing three fundamental questions: 1) where and how frequently have global coastal oceans been affected by phytoplankton blooms? 2) have the blooms expanded or intensified over the past two decades, both globally and regionally? and 3) what are the potential drivers? |
| Research sample | Three separate samples were selected. 1) MODIS Aqua images were used to develop the phytoplankton bloom extraction algorithm, 2) MODIS Aqua images and were used to verify the reliability of the algorithm and the accuracy of the phytoplankton bloom extraction results, and 3) in situ reported HAB events from the HAEDAT dataset were used to validate the accuracy of the phytoplankton bloom extraction results. |
| Sampling strategy | A total of 115 MODIS Aqua images were selected from the different locations where coastal phytoplankton blooms have been recorded in the published literature, of which 80 were used for algorithm development and 35 were used for algorithm validation. A total number of 2609 HAB events that occurred in the coastal area were selected from the HAEDAT dataset. |
| Data collection | The HAEDAT dataset is a collection of records of harmful algal bloom (HAB) events , maintained under the UNESCO Intergovernmental Oceanographic Commission and with data archives since 1985. |
| Timing and spatial scale | The satellite data were acquired from different seasons and across various phytoplankton bloom magnitudes between 2003 and 2020, and HAB data from 2003 to 2020 in the HAEDAT dataset were used. |
| Data exclusions | No data were excluded from analysis. |
| Reproducibility | Our results could easily be reproduced with existing datasets. |
| Randomization | Excluding data affected by clouds, a total of 0.76 million MODIS Aqua images from 2003 to 2020 were used to extract phytoplankton blooms in global coastal area. |
| Blinding | Not applicable in our study. |

Did the study involve field work?  ☐ Yes  ☒ No

# Reporting for specific materials, systems and methods

We require information from authors about some types of materials, experimental systems and methods used in many studies. Here, indicate whether each material, system or method listed is relevant to your study. If you are not sure if a list item applies to your research, read the appropriate section before selecting a response.

## Materials & experimental systems

| n/a | Involved in the study |
|-----|-----------------------|
| ☒ ☐ | Antibodies |
| ☒ ☐ | Eukaryotic cell lines |
| ☒ ☐ | Palaeontology and archaeology |
| ☒ ☐ | Animals and other organisms |
| ☒ ☐ | Clinical data |
| ☒ ☐ | Dual use research of concern |

## Methods

| n/a | Involved in the study |
|-----|-----------------------|
| ☒ ☐ | ChIP-seq |
| ☒ ☐ | Flow cytometry |
| ☒ ☐ | MRI-based neuroimaging |

