## [Peer Review File · Nature]

Manuscript Title: Coastal phytoplankton blooms expand and intensify in the 21st 1 century

Reviewer Comments & Author Rebuttals

Reviewer Reports on the Initial Version:

Referees' comments:

Referee #1 (Remarks to the Author):

It is a very well-written paper on intensifying coastal phytoplankton blooms using ocean colour data from the MODIS Aqua satellite. The paper uses a broad range of data and models to examine the dependence of excessive phytoplankton growth on sea surface temperature. A very important new contribution to the problems is the development of an automated bloom detection method to examine the dependence on climate change and human activities, including changes in SST and SST gradient and climate extremes. I have the SST background so I only have two minor comments (which I leave to the authors to decide on how to address).

I understand why the authors use the National Oceanic and Atmospheric Administration Optimum Interpolated SST (OISST) data to examine the effects of warming on the global phytoplankton trends. It integrates the instrumental record with model forecasts and produces a coherent set of SST for the analysis. However, lots of the phytoplankton blooms are in the coastal oceans; as indicated by the method part, the bloom count was averaged over all 1-km pixels using L3 MODIS data. And the (OISST) resolution is 25km; it also has high uncertainties in coastal waters. How do you handle this difference? Since MODIS also has the same resolution L2 and L3 SST data. Whether we can get better results with the simultaneously observed data from the same sensor?

You have the unique HAEDAT dataset and accurate classification algorithms; it may be presented in more illuminating ways to illustrate clearly the major functional dependencies of the phytoplankton blooms compared to other aspects of the climate variables such as precipitation, wind speed, etc. over the coastal regions. You have a gem here, but haven't polished it.

Scientifically, I would say that this is a well-done work that certainly deserves to be published in Nature.

Bingkun Luo
Smithsonian Center for Astrophysics
Harvard University

Referee #2 (Remarks to the Author):

This paper is very well written and well structured, and the methods and results are clear. It is a novel contribution that should be published.

The work assesses trends in frequency and spatial extent of global coastal algal blooms over an 18-year period from 2003-2020. It uses a newly developed bloom detection method as applied to observed top-of-atmosphere reflectance measured by the MODIS sensor on the Aqua satellite, after subtraction of the dominant Rayleigh reflectance of the atmosphere. Over the study period, the authors find that the frequency and spatial extent of algal blooms increased significantly in most of the world's coastal waters (as defined by the international boundaries of economic exclusion zones).

I found only minor issues in my review.

The use of the term "bloom intensity" through-out the manuscript is somewhat misleading. Would it not be more appropriate to say "bloom frequency"?

While the detection algorithm is able to distinguish between sediments and phytoplankton blooms, does the presence of suspended sediments reduce the efficacy of the detection algorithm, as in Gilerson et al. 2008, <https://doi.org/10.1364/OE.16.002446>? If so, please note the potential limitation. Perhaps this contributes to the discrepancies with HAEDAT.

While the bloom detection approach is new, it would be appropriate to note that the use of Rrc as an alternative to Rrs in complex waters is not novel. e.g., Feng et al. 2018, 10.1016/j.isprsjprs.2018.08.020

19: Suggest to temper or clarify this first line of the abstract, since you correctly note later in the manuscript that not all increases in productivity are a problem.

23: Does this cover all known coastal EEZs. Are there more countries with coastlines than the 153 coastal countries examined? Please clarify.

70: Is 760,000 the number of 5-minute MODIS/Aqua granules that touch one or more EEZs? Please clarify.

173: is the anti-correlation with SST in these regions due to stratification and thus reduced nutrient availability?

234 (and 256): extraneous comma.

402: suggest to say downloaded from the Ocean Biology Distributed Active Archive Center (OB.DAAC) at NASA Goddard Space Flight Center

404: just noting that this is an old version of SeaDAS. Is there a reason for using the older version?

404: I don't think there is a product called Rrc that can be directly output from SeaDAS. Please clarify in methods: did you output toa radiance and rayleigh radiance and compute Rrc, or is this rhos

(quasi-surface reflectance)?

407: The straylight flag is often triggered by land adjacency. Did you look at the impact of this flag on your statistics? Might it be a factor in the reduced bloom frequency of your method relative to HAEDAT.

409: Were the AOT data obtained from the OB.DAAC? Version R2018.0? Please clarify data sources.

416: suggest to rephrase: "of marine resources available for human use".

454: suggest to rephrase: "a proxy for phytoplankton biomass".

729: extraneous word "in"

Referee #3 (Remarks to the Author):

The current manuscript uses a new satellite ocean color algorithm to detect phytoplankton blooms globally across exclusive economic zones. The analysis is thorough, identifies important trends in bloom intensity and area, and relates these bloom changes to environmental drivers. The manuscript is very well written and interesting. I only have a few very minor editorial corrections/suggestions.

1. (last word on line 103) Change 'in' to 'around'
2. (line 104) Define 'oceania'. This may be familiar to oceanographers, but Nature has a very broad readership and some might not be familiar with this term.
3. (general) Do a global document search for semicolons (;) and evaluate if the sentence might be clearer if split into two sentences. I believe in many cases it will be.
4. (extended figure 1) define the hatched black boxes in the 3 lower panels
5. (extended figures 3, 4, 5, 6) define the coloring in the bloom area detected figures. For example, in ED-Fig 3, "...and the bloom area detected (green pixels) by the..."
6. (extended figure 4) the title of this figure seems misleading, as the reader expects to see a comparison of Rrc and Rrs approaches. I think it would be better written as something like "Examples showing disadvantages of using Rrs (...) in algal bloom detection."
7. (extended figure 5) provide a clearer description of each column of images in the figure legend for panel a.
8. (extended figure 6) revise on line 41 to read, "...are required for (labeled at bottom) RI, ABI..."

9. (extended figure 9) please define 'bond curves'. I didn't know what this meant and I suspect others will not either.

10. (extended figure 10) the way this caption currently reads it implies that panel d show trends in global fertilizer used for fisheries production. Is that correct??

Referee #4 (Remarks to the Author):

General comments: As stated in the abstract, the goals of this manuscript are to map coastal phytoplankton blooms during the period between 2003 and 2020 using satellite imagery at a spatial resolution of 1 km. The authors developed an improved algorithm to achieve this goal and processed more than 0.7 million images to generate the global maps of algal bloom distribution. The processed observations were then used to look at trends over the 17-year period covering the study. The authors found a significant increase in coastal blooms globally and presented their results also in a breakdown for 153 coastal countries. The authors then attempt to link the increase in phytoplankton blooms to increases in sea surface temperature and anthropogenic nutrient enhancement, two of the most obvious causes for these increase in frequency and intensity of the blooms. The authors further argue that their analysis can provide the basis for risk assessment, management and policy actions.

The authors present an extensive method section that explains the design and validation of their algorithm and the subsequent acquisition of bloom detection. There is extensive discussion of how error from cloud was treated, however, it would be important to provide information on how many images were discarded in the process and whether the removal of these images generated a seasonal or spatial bias the overall data acquisition. It is possible that I missed it however, there a need to provide a more transparent account of the total number of images inspected and discarded to assess whether temporal or spatial bias was generated in the sampling.

The presentation of the data by continent and the more detail account at the country level is questionable in my opinion because there are major ocean circulation factors that are responsible for some of the observed patterns, as correctly pointed out by the authors. However, the distinction of blooms per country can give the wrong perception. For example, in supplemental table 2, all the Baltic states bordering the Baltic Sea have the same overall bloom incidence and spatial covering, so I do not see the need to report the data by country, when it should be more by regional coastal areas for enclosed seas. The authors also attempt to find causal relationships with ocean currents, fertilizer input and fisheries aquaculture. While the association of high bloom incidence with eastern boundary currents is obvious, and also already well established, the connection to the nutrient input is not well supported by the data presented in this manuscript. Moreover, the sources of data and information used to generate extended Figure 10 are not clearly documented.

Overall there is a dichotomy of ideas and concepts in the manuscript that need to be focused. The title indicates that the work is focused on coastal phytoplankton blooms, and then looking at the generated maps, the data includes the North Sea, Western North Atlantic, and additional offshore areas. Extended Figure 9 for example.

Specific comments:

Abstract: The goals are over-reaching in consideration with the analyses, especially with the claim of linking to increase nutrient input and aquaculture efforts.

Line 115: explain why this graph is significant in terms of the size of the coastline. Especially why you would expect a meaningful relationship between the two variables.

Line 124: Estonia, Lithuania, Latvia, Poland, Sweden, Finland are all bordering the Baltic Sea

Line 145: Some error or uncertainty assessment on the individual yearly estimates are needed for Figure 3b

Line 259: Our daily mapping of bloom events offers critical insight into the mechanisms underlying the formation, maintenance, and dissipation of algal blooms. Many of the points discussed in this manuscript are already known, like the association of blooms with ocean currents and upwelling regions. The authors have not communicated what additional insight they provide at the global scale.

Line 414 <https://www.marineregions.org/downloads.php>: this web site is too generic: How exactly were the EEZ calculated? There are several links on this website that discuss EEX

Line 447: Remarking on the increase of HAB: The authors have not differentiated their data into HAB and phytoplankton bloom that are part of the general oceanic productivity patterns, for those in the North Sea and upwelling regions.

Line 449: However, such an overall increasing trend was found to be highly correlated with recently intensified sampling efforts. The authors do not explain how the correction of the bias was obtained.

Line 569: Depending on the local region and application purpose, the meaning of “phytoplankton bloom” may be different. The definition of a phytoplankton bloom is unclear. Are the blooms identified by a threshold? Relative to the background in an area?

Extended figure 10: The data on the fertilizer as per the provided link (Line 438) goes to 2019. It is not clear how the maps were generated and why the data is only up to 2013. The colour scales on Figure 10 are also misleading starting at -2000 tonnes per year for aquaculture. It should start at 0, and go much higher to account for the difference in production for China and Brazil, as per the data on the websites provided by the authors.

Extended data: Figure 9 there is no (f).

Author Rebuttals to Initial Comments:

Referee #1 (Remarks to the Author):

General remarks

It is a very well-written paper on intensifying coastal phytoplankton blooms using ocean colour data from the MODIS Aqua satellite. The paper uses a broad range of data and models to examine the dependence of excessive phytoplankton growth on sea surface temperature. A very important new contribution to the problems is the development of an automated bloom detection method to examine the dependence on climate change and human activities, including changes in SST and SST gradient and climate extremes. I have the SST background so I only have two minor comments (which I leave to the authors to decide on how to address).

Reply: Thank you for your encouraging comments.

I understand why the authors use the National Oceanic and Atmospheric Administration Optimum Interpolated SST (OISST) data to examine the effects of warming on the global phytoplankton trends. It integrates the instrumental record with model forecasts and produces a coherent set of SST for the analysis. However, lots of the phytoplankton blooms are in the coastal oceans; as indicated by the method part, the bloom count was averaged over all 1-km pixels using L3 MODIS data. And the (OISST) resolution is 25km; it also has high uncertainties in coastal waters. How do you handle this difference? Since MODIS also has the same resolution L2 and L3 SST data. Whether we can get better results with the simultaneously observed data from the same sensor?

Reply: Thanks for pointing out this issue. We agree that MODIS has higher resolution (4 km) SST products (<https://oceancolor.gsfc.nasa.gov/l3/>), using the algorithm developed by Dr. Peter Minnett's group. However, the use of 25 km does not cause significant differences in our analysis, which were based on either the spatial consistency between the global SST (or SST gradient) and bloom patterns or the correlations of their mean values over a large current system. For example, we compared the time series of annual mean SST between different resolutions for two current systems, where the data show very similar trends (see below). As 25 km resolution data for both SST and SST gradient are readily available through Castaneda-Guzman, et al. ¹), we simply obtained their data for our analysis.

References

1 Castaneda-Guzman, M., Mantilla-Saltos, G., Murray, K. A., Settlage, R. & Escobar, L. E. A database of global coastal conditions. *Scientific Data* **8**, 304, doi:10.1038/s41597-021-01081-9 (2021).

You have the unique HAEDAT dataset and accurate classification algorithms; it may be presented in more illuminating ways to illustrate clearly the major functional dependencies of the phytoplankton blooms compared to other aspects of the climate variables such as precipitation, wind speed, etc. over the coastal regions. You have a gem here, but haven't polished it.

Reply: Thanks for this thoughtful comment. Yes, the major contribution of our work is to provide the first global map of coastal phytoplankton blooms. We believe more in-depth analysis using our data could help to understand the mechanisms underlying the formation, maintenance, and dissipation of algal blooms; we have discussed this point in our manuscript. However, for your reference, we have performed the analysis using precipitation and wind speed datasets (see below), which showed weaker correlations than SST or SST gradients. Again, we would like to emphasize that we will open our globally mapped bloom dataset to the public, and seek further collaborations with ecologists and oceanographers to understand the global trends revealed from satellite observations.

Scientifically, I would say that this is a well-done work that certainly deserves to be published in Nature.

Reply: Thank you so much for your recommendation.

Referee #2 (Remarks to the Author):

General remarks

This paper is very well written and well structured, and the methods and results are clear. It is a novel contribution that should be published.

The work assesses trends in frequency and spatial extent of global coastal algal blooms over an 18-year period from 2003-2020. It uses a newly developed bloom detection method as applied to observed top-of-atmosphere reflectance measured by the MODIS sensor on the Aqua satellite, after subtraction of the dominant Rayleigh reflectance of the atmosphere. Over the study period, the authors find that the frequency and spatial extent of algal blooms increased significantly in most of the world's coastal waters (as defined by the international boundaries of economic exclusion zones). I found only minor issues in my review.

Reply: Thank you for your encouraging recommendation. We have made itemized changes to address your comments.

Specific remarks

Comment 1. The use of the term "bloom intensity" through-out the manuscript is somewhat misleading. Would it not be more appropriate to say "bloom frequency"?

Reply: We have changed it into "**bloom frequency**" throughout the manuscript.

Comment 2. While the detection algorithm is able to distinguish between sediments and phytoplankton blooms, does the presence of suspended sediments reduce the efficacy of the detection algorithm, as in Gilerson et al. 2008, <https://doi.org/10.1364/OE.16.002446>? If so, please note the potential limitation. Perhaps this contributes to the discrepancies with HAEDAT.

Reply: When blooms occur in highly turbid waters, this could be a possible cause to the discrepancies with HAEDAT. However, under normal conditions, the growth of phytoplankton should be restricted with the presence of suspended sediments, by blocking sunlight and other factors. At the current stage, we are not able to quantify the associated impacts, and we acknowledged this point in the revision: "**other types of HAB events in the HAEDAT dataset may not have been detectable by satellite observations, such as events with lower phytoplankton biomass but high toxicity, occurrences at the subsurface layers, or fluorescence signals overwhelmed by suspended sediments** ⁶⁸⁻⁷⁰ (see line 679 to 682).", and with your suggested reference added.

Comment 3. While the bloom detection approach is new, it would be appropriate to note that the use of Rrc as an alternative to Rrs in complex waters is not novel. e.g., Feng et al. 2018, [10.1016/j.isprsjprs.2018.08.020](https://doi.org/10.1016/j.isprsjprs.2018.08.020)

Reply: Yes, the paper you mentioned is also from our group, we have acknowledged the use of R_{rc} is not novel in the revision as **“In fact, R_{rc} has been used as an alternative to R_{rs} in various applications in complex waters^{60, 61}”** (see line 560 to 561).

Comment 4. 19: Suggest to temper or clarify this first line of the abstract, since you correctly note later in the manuscript that not all increases in productivity are a problem.

Reply: We have rephrased this sentence as **“Excessive phytoplankton growth, or blooms, in coastal oceans, can be beneficial to coastal fisheries production and ecosystem function, but also can cause major environmental problems^{1,2}, yet detailed characterizations of bloom incidence and distribution are unavailable worldwide.”**

Comment 5. 23: Does this cover all known coastal EEZs. Are there more countries with coastlines than the 153 coastal countries examined? Please clarify.

Reply: We further clarified in the data source section (lines 414 to 417): **“We examined the algal blooms in the Exclusive Economic Zones (EEZs) of 153 ocean-bordering countries (excluding the EEZs in the Caspian Sea or around the Antarctic), 126 of which were found with at least one bloom in the past two decades.”**

Comment 6. 70: Is 760,000 the number of 5-minute MODIS/Aqua granules that touch one or more EEZs? Please clarify.

Reply: In fact, we applied our algorithm to all MODIS Aqua images within our examined period of 2003-2020, with a total of 760,000 5-minute granules. Then, to perform global or regional statistics, the data outside our study regions (i.e., EEZs) were excluded. We clarified in the revision (line 71 to 74) as **“The dataset was derived using global, 1-km resolution daily observations from the Moderate Resolution Imaging Spectroradiometer (MODIS) onboard the NASA’s Aqua satellite, and all 0.76 million images acquired by this satellite mission between 2003 and 2020 were used”**.

Comment 7. 173: is the anti-correlation with SST in these regions due to stratification and thus reduced nutrient availability?

Reply: Yes, the increase in SST could strengthen the stratification of the oceans and reduce nutrient availability. While quantifying such effects appears challenging at the current stage and beyond the scope of our study (whose main objective is to provide the global patterns and trends of blooms), we would like to contribute to the related studies by helping and collaborating with other groups after our datasets have been released to the public.

To address your comment, we revised our texts in lines 173 to 175: **“Climate changes can also affect ocean circulation, altering ocean mixing and the transport of nutrients that drive the growth of marine phytoplankton and bloom formation”**.

We have also acknowledged in our manuscript: **“an ecosystem model that incorporates terrestrial and oceanic nutrient transport and nutrient-plankton relationships of different**

species, is required to quantify the contributions of natural and anthropogenic factors to algal blooms.”

Comment 8. 234 (and 256): extraneous comma.

Reply: We have removed all unnecessary semicolons throughout the manuscript to address this comment.

Comment 9. 402: suggest to say downloaded from the Ocean Biology Distributed Active Archive Center (OB.DAAC) at NASA Goddard Space Flight Center

Reply: Changed as suggested.

Comment 10. 404: just noting that this is an old version of SeaDAS. Is there a reason for using the older version?

Reply: Yes, the latest version of SeaDAS is 8.1, and the major update over SeaDAS 7.5 is the GUI. However, the atmospheric correction process was based on the OCSSW component, which was not updated for MODIS processing.

Comment 11. 404: I don't think there is a product called Rrc that can be directly output from SeaDAS. Please clarify in methods: did you output toa radiance and rayleigh radiance and compute Rrc, or is this rhos (quasi-surface reflectance)?

Reply: It was converted using the rhos (in sr^{-1}) product ($rhos \times \pi$) from SeaDAS, and we have clarified this in the revision.

Comment 12. 407: The straylight flag is often triggered by land adjacency. Did you look at the impact of this flag on your statistics? Might it be a factor in the reduced bloom frequency of your method relative to HAEDAT.

Reply: Yes, the straylight flag near land is designed to avoid land adjacency effects on the satellite signal; when a pixel is classified as land, the 7*5 pixels centered at this pixel would be flagged as straylight. We have acknowledged this point in the revision as: **“Fourth, a reduced MODIS satellite observation frequency by the contaminations of clouds and land adjacency effects”**⁷²

Comment 13. 409: Were the AOT data obtained from the OB.DAAC? Version R2018.0? Please clarify data sources.

Reply: Yes, AOT data were also obtained from the OB.DAAC NASA GSFC (Version R2018.0), we clarified in the revision.

Comment 14. 416: suggest to rephrase: "of marine resources available for human use".

Reply: Changed as suggested.

Comment 15. 454: suggest to rephrase: "a proxy for phytoplankton biomass".

Reply: Changed as suggested.

Comment 16. 729: extraneous word "in"

Reply: Deleted "in".

Referee #3 (Remarks to the Author):

General remarks

The current manuscript uses a new satellite ocean color algorithm to detect phytoplankton blooms globally across exclusive economic zones. The analysis is thorough, identifies important trends in bloom intensity and area, and relates these bloom changes to environmental drivers. The manuscript is very well written and interesting. I only have a few very minor editorial corrections/suggestions.

Reply: Thank you for your encouraging comments.

Specific remarks

Comment 1. (last word on line 103) Change 'in' to 'around'

Reply: Changed as suggested.

Comment 2. (line 104) Define 'oceania'. This may be familiar to oceanographers, but Nature has a very broad readership and some might not be familiar with this term.

Reply: Thank you for this suggestion. We have changed "oceania" into "Australia" throughout the manuscript.

Comment 3. (general) Do a global document search for semicolons (;) and evaluate if the sentence might be clearer if split into two sentences. I believe in many cases it will be.

Reply: We have removed all unnecessary semicolons throughout the manuscript to address this comment, as also suggested by Referee #2.

Comment 4. (extended figure 1) define the hatched black boxes in the 3 lower panels

Reply: The hatched black boxes in the three lower panels highlight different spectral shapes between phytoplankton blooms and other features near the fluorescence band. We have added this information in the revision.

Comment 5. (extended figures 3, 4, 5, 6) define the coloring in the bloom area detected figures. For example, in ED-Fig 3, "...and the bloom area detected (green pixels) by the..."

Reply: The revision has added the definition of the bloom areas for these figures.

Comment 6. (extended figure 4) the title of this figure seems misleading, as the reader expects to see a comparison of R_{rc} and R_{rs} approaches. I think it would be better written as something like “Examples showing disadvantages of using R_{rs} (...) in algal bloom detection.”

Reply: Thanks for this suggestion. The caption of this figure has been written as “**Examples showing disadvantages of using NASA standard R_{rs} (i.e., with the removal of both Rayleigh and aerosol scattering) in algal bloom detection. From left to right are the RGB composites, ERGB, nFLH, and the bloom areas (green pixels) detected by the CIE-fluorescence algorithm (based on R_{rc} , without the removal of aerosol scattering).**”

Comment 7. (extended figure 5) provide a clearer description of each column of images in the figure legend for panel a.

Reply: The left columns show the RGB composites, and the right three columns show the bloom areas under different AOTs. We have added this information to the revised figure caption.

Comment 8. (extended figure 6) revise on line 41 to read, “...are required for (labeled at bottom) RI, ABI...”

Reply: Should be “(labeled within the panels)”, we have added in the revision.

Comment 9. (extended figure 9) please define ‘bond curves’. I didn’t know what this meant and I suspect others will not either.

Reply: We have clarified as “**colored bond curves**”.

Comment 10. (extended figure 10) the way this caption currently reads it implies that panel d show trends in global fertilizer used for fisheries production. Is that correct??

Reply: Thanks for catching this point. We have revised the caption to avoid confusion as “**(b-c) Trends of nitrogen and phosphorus from 2003 to 2019 for different countries. (d) Trends of fishery production from 2003 to 2018.**”

Referee #4 (Remarks to the Author):

General remarks

General comments: As stated in the abstract, the goals of this manuscript are to map coastal phytoplankton blooms during the period between 2003 and 2020 using satellite imagery at a spatial resolution of 1 km. The authors developed an improved algorithm to achieve this goal and processed more than 0.7 million images to generate the global maps of algal bloom distribution. The processed observations were then used to look at trends over the 17-year period covering the study. The authors found a significant increase in coastal blooms globally and presented their results also in a breakdown for 153 coastal countries. The authors then attempt to link the increase in phytoplankton blooms to increases in sea

surface temperature and anthropogenic nutrient enhancement, two of the most obvious causes for these increase in frequency and intensity of the blooms. The authors further argue that their analysis can provide the basis for risk assessment, management and policy actions.

Reply: Thank you for your thoughtful comments.

The authors present an extensive method section that explains the design and validation of their algorithm and the subsequent acquisition of bloom detection. There is extensive discussion of how error from cloud was treated, however, it would be important to provide information on how many images were discarded in the process and whether the removal of these images generated a seasonal or spatial bias the overall data acquisition. It is possible that I missed it however, there a need to provide a more transparent account of the total number of images inspected and discarded to assess whether temporal or spatial bias was generated in the sampling.

Reply: Good point. We have added a figure (Supplementary Fig. 1) to show the number of valid pixel observations (N_{Vobs}) on annual and quarterly scales, and also examined their long-term patterns. Results show that the mean annual N_{Vobs} for global $1^\circ \times 1^\circ$ grid cells is $\sim 2.0 \times 10^5$, and the fluctuation patterns and trends, either annually or seasonally, are different from that of the global bloom frequency and affected areas. We have added related information in the revision (see lines 620 to 624).

Supplementary Fig. 1 | The number of valid pixel observations (N_{Vobs}) for global $1^\circ \times 1^\circ$ grid cells. (a) Global pattern of mean annual N_{Vobs} , and the mean values for four quarters are shown from c-f (JFM: January, February, and March, AMJ: April, May, and June, JAS: July,

August, and September, OND: October, November, and December). The interannual patterns from 2003 and 2020 are demonstrated in (b).

The presentation of the data by continent and the more detail account at the country level is questionable in my opinion because there are major ocean circulation factors that are responsible for some of the observed patterns, as correctly pointed out by the authors. However, the distinction of blooms per country can give the wrong perception. For example, in supplemental table 2, all the Baltic states bordering the Baltic Sea have the same overall bloom incidence and spatial covering, so I do not see the need to report the data by country, when it should be more by regional coastal areas for enclosed seas. The authors also attempt to find causal relationships with ocean currents, fertilizer input and fisheries aquaculture. While the association of high bloom incidence with eastern boundary currents is obvious, and also already well established, the connection to the nutrient input is not well supported by the data presented in this manuscript. Moreover, the sources of data and information used to generate extended Figure 10 are not clearly documented.

Reply: To address your recommendation on the statistics of using “regional coastal areas for enclosed seas”, we have added the analysis of large marine ecosystems (LMEs) in the revision. The LMEs are relatively large regions on the order of at least 200,000 km², which are characterized by distinct bathymetry, hydrography, productivity, and trophically dependent populations. We have added a figure (Extended Data Fig. 7), a table (Supplementary Table 3), and related texts (lines 116 to 126) to present the statistics for different LMEs. The Baltic Sea you mentioned ranked third in annual mean bloom count. Also, we have kept the country-based statistics in the manuscript, as the nutrients and aquaculture datasets used in the further analysis are provided at the country level. More importantly, we believe some potential readers from the management sectors (given the high impact of Nature) would be interested in the statistics for different countries.

Extended Data Fig. 7 | Annual median bloom count and the proportion of bloom-affected areas for large marine ecosystems (LMEs). (a) Annual median bloom count, (b) proportion of bloom-affected areas. The data are ordered from the largest to the smallest. The LMEs are grouped by continent, and their names, numbers, and locations are shown in (a) and (b).

To address your concerns about the analysis of nutrient input, we have removed the related conclusion in the abstract. Indeed, we have also acknowledged in the text that **“an ecosystem model incorporating terrestrial and oceanic nutrient transport and nutrient-plankton relationships of different species³⁹ is required to quantify the contributions of natural and anthropogenic factors to algal blooms¹⁴.”**

We are sorry for the confusion caused by the caption of Extended Figure 10, and have now been corrected (please see specific reply below).

We further emphasized the point in the revision that **“The major contribution of our study is to provide the first spatially and temporally consistent characterization of global coastal algal blooms between 2003 and 2020”**, and we have changed our statement on the related analysis, and claimed that **“Our daily mapping of bloom events offers critical baseline information to understand the mechanisms underlying the formation, maintenance, and dissipation of algal blooms.”**

Overall there is a dichotomy of ideas and concepts in the manuscript that need to be focused. The title indicates that the work is focused on coastal phytoplankton blooms, and then looking at the generated maps, the data includes the North Sea, Western North Atlantic, and additional offshore areas. Extended Figure 9 for example.

Reply: Thank you for pointing out this issue. When performing global and regional statistics of the blooms in “coastal ocean”, we tried to find an “accurate” definition for “coastal”. However, it appears that “coastal waters” is a rather general term used differently in different contexts. Nevertheless, we found two definitions that could more comprehensively include the global coastal conditions: 1) exclusive economic zones (EEZs), which are marine zones within 200 nautical miles from a country’s coastline where each country claims jurisdiction for economic activities (see examples in Castaneda-Guzman, et al. ¹, Halpern, et al. ²). and 2) large marine ecosystems (LMEs), which encompass coastal areas from river basins and estuaries to the seaward boundaries of continental shelves, enclosed and semi-enclosed seas, and the outer margins of the major current systems. They are relatively large regions on the order of at least 200,000 km², characterized by distinct bathymetry, hydrography, productivity, and trophically dependent populations (see example in Guo, et al. ³). In fact, the boundaries of LMEs and EEZs are exactly the same for some regions (such as Australia, the North Sea) and could differ in others, whereas they both cover your mentioned areas (North Sea and Western North Atlantic, see the figure in above reply). Apparently, the two definitions are of different foci of interest, and we used EEZs in our previous submission. However, the global trends of using the two different definitions resulted in similar global trends (see figure below).

Fig. S1. Global trends of bloom frequency using the two different boundaries of coastal oceans. Blue: EEZ, Red: LME.

To further address your concerns, we made the following changes in the revision:

- 1) We included the analysis of using LMEs (see reply above).
- 2) We further clarified the examined regions in our study in both the main text and the method section.

Lines 88 to 93: **We examined phytoplankton blooms in the exclusive economic zones of 153 coastal countries and in 54 large marine ecosystems (LMEs) (Supplementary Table 3, Extended Data Fig. 7). Our study area encompasses global continental shelves and outer margins of coastal currents, which offer the majority of marine resources available for human use.**

Lines 414 to 428: **We examined the algal blooms in the Exclusive Economic Zones (EEZs) of 153 ocean-bordering countries (excluding the EEZs in the Caspian Sea or around the Antarctic), 126 of which were found with at least one bloom in the past two decades. The EEZs dataset is available at https://www.marinerregions.org/download_file.php?name=World_EEZ_v11_20191118.zip. The EEZs are up to 200 nautical miles (or 370 km) away from coastlines, which include all continental shelf areas and offer the majority of marine resources available for human use. Regional statistics of algal blooms were also performed for LMEs. LMEs encompass global coastal oceans and outer edges of coastal currents areas, which are defined by various distinct features of the oceans, including hydrology, productivity, bathymetry, and trophically dependent populations⁴. Of the 66 LMEs identified globally, we excluded the Arctic and Antarctic regions and examined a total of 54 LMEs. The boundaries of LMEs were obtained from <https://www.sciencebase.gov/catalog/item/55c77722e4b08400b1fd8244>.**

References

- 1 Castaneda-Guzman, M., Mantilla-Saltos, G., Murray, K. A., Settlage, R. & Escobar, L. E. A database of global coastal conditions. *Scientific Data* **8**, 304, doi:10.1038/s41597-021-01081-9 (2021).
- 2 Halpern, B. S. *et al.* An index to assess the health and benefits of the global ocean. *Nature* **488**, 615-620, doi:10.1038/nature11397 (2012).

3 Guo, X. *et al.* Threat by marine heatwaves to adaptive large marine ecosystems in an eddy-resolving model. *Nature Climate Change* **12**, 179-186, doi:10.1038/s41558-021-01266-5 (2022).

4 Sherman, K. Adaptive management institutions at the regional level: The case of Large Marine Ecosystems. *Ocean & Coastal Management* **90**, 38-49, doi:<https://doi.org/10.1016/j.ocecoaman.2013.06.008> (2014).

Specific remarks

Comment 1. Abstract: The goals are over-reaching in consideration with the analyses, especially with the claim of linking to increase nutrient input and aquaculture efforts.

Reply: Thanks for pointing out this issue. We have removed the claim on nutrients from the abstract, and the related sentence has been changed into “**We documented the relationship between the bloom trends and ocean circulation, and identified the stimulatory effects of recent increases in sea surface temperature**”.

Comment 2. Line 115: explain why this graph is significant in terms of the size of the coastline. Especially why you would expect a meaningful relationship between the two variables.

Reply: Very good point. The previous Fig. 2 and related correlation analysis has been removed to address your comment.

Comment 3. Line 124: Estonia, Lithuania, Latvia, Poland, Sweden, Finland are all bordering the Baltic Sea

Reply: We have added the analysis for LMEs, and the Baltic Sea is one of the LMEs, where the detailed bloom information has been presented in Extended Data Fig. 7 and Supplementary Table 3. Please also see the reply above for detailed changes in the revision.

Comment 4. Line 145: Some error or uncertainty assessment on the individual yearly estimates are needed for Figure 3b.

Reply: Good point. As the overall accuracy of our satellite bloom detection algorithm is 95% (Supplementary Table 1), we have added 5% of errors into the annual median bloom frequency and shaded them into Figure 2b (Figure 3b in the previous submission). In addition to the global annual median values shown in Figure 2b, we further examined the trends for 25% and 75% percentiles of the global bloom frequency, which also resulted in similar increasing trends (the results are not shown in this figure due to the large range of global values).

The bloom-affected area in Figure 2b was defined as the areas where algal blooms were detected at least once by satellite observations, representing the maximum possible bloom areas. Therefore, further quantification of the associated errors is difficult, and we believe the magnitudes should be much smaller than 5%.

Figure 2b. Interannual variability and trends of annual median bloom frequency (left axis) and total global bloom-affected areas (right axis), and their linear slopes and significance are annotated. The colors of the curves correspond to the y-axes. *The shading associated with bloom frequency represents an uncertainty level of 5% in bloom detection.*

Comment 5. Line 259: Our daily mapping of bloom events offers critical insight into the mechanisms underlying the formation, maintenance, and dissipation of algal blooms. Many of the points discussed in this manuscript are already known, like the association of blooms with ocean currents and upwelling regions. The authors have not communicated what additional insight they provide at the global scale.

Reply: We have further highlighted in the conclusion that *“The major contribution of our study is to provide the first spatially and temporally consistent characterization of global coastal algal blooms between 2003 and 2020”*. And we further adjusted the text of your mentioned statements as *“Our daily mapping of bloom events offers critical baseline information to understand the mechanisms underlying the formation, maintenance, and dissipation of algal blooms. This could aid in the development of forecasting models (on either global or regional scales) that can help to minimize the consequences of harmful blooms, and can also help in policy decisions relating to the control of nutrient discharges and other HAB-stimulatory factors.”*

Comment 6. Line 414 <https://www.marineregions.org/downloads.php>: this web site is too generic: How exactly were the EEZ calculated? There are several links on this website that discuss EEX.

Reply: By definition, the EEZs are areas adjacent to the country's territorial seas, and with extensions of <200 nautical miles out from their coastlines. We have updated the data link: https://www.marineregions.org/download_file.php?name=World_EEZ_v11_20191118.zip. However, how these boundaries were created was not listed on their website, and we only find the related information here: <https://www.marineregions.org/eez.php>. As mentioned in previous replies, the concept of EEZ has been used by many studies to perform global-scale coastal ocean studies, and therefore we adopted the concept in our study.

Comment 7. Line 447: Remarking on the increase of HAB: The authors have not differentiated their data into HAB and phytoplankton bloom that are part of the general oceanic productivity patterns, for those in the North Sea and upwelling regions.

Reply: Yes, we agree that current optical satellite missions are not possible to differentiate HAB and other phytoplankton blooms. Indeed, we have mentioned this point in several places in our paper. A few examples are provided below.

Line 77 to 79 : **“However, whether a bloom produces toxins or harms humans or the marine environment is not distinguishable from satellites.”**

Line 263 to 266: **“Noting again that many blooms are beneficial, particularly in terms of their positive impacts on ecosystems as well as wild and farmed fisheries, the results here can also contribute toward policies and management actions that sustain those beneficial blooms.”**

Line 713 to 715: **“We acknowledge that our satellite-detected algal blooms represent only high amounts of phytoplankton biomass on the ocean surfaces without distinguishing whether such blooms produce toxins or are harmful to marine environments.”**

And we further clarified this point in the abstract:

Line 21 to 23: **“Excessive phytoplankton growth, or blooms, in coastal oceans, can *be beneficial to coastal fisheries production* and ecosystem function, but also can cause major environmental problems”**

Line 31 to 34: **“Our compilation of daily mapped coastal phytoplankton blooms provides the basis for global assessments of bloom risks *and benefits*, and for the formulation or evaluation of management or policy actions”**

Comment 8. Line 449: However, such an overall increasing trend was found to be highly correlated with recently intensified sampling efforts. The authors do not explain how the correction of the bias was obtained.

Reply: The correction was performed by the original study instead of us. We have revised the text to further clarify their method with the reference: **“Once this potential bias was accounted for *by examining the ratio between HAB events to the number of samplings*⁵, there was no significant global trend in HAB incidence, though there were increases in certain regions”**.

Comment 9. Line 569: Depending on the local region and application purpose, the meaning of “phytoplankton bloom” may be different. The definition of a phytoplankton bloom is unclear. Are the blooms identified by a threshold? Relative to the background in an area?

Reply: Not really. The bloom was not identified by a threshold. For each pixel, we estimate the xy coordinate in the CIE color space, and it will be classified as the bloom when the coordinate is located above the lower boundary of the sample points (lower bound of the fluorescence signal that can be detected by MODIS). We further clarified this point in the revision (lines 587 to 591): **“Instead of a simple threshold, we used a lower boundary of the sample points in the chromaticity diagram to define bloom. In simple words, a pixel is**

classified as a bloom if its fluorescence signal is detectable (i.e., the associated xy coordinate in the CIE color space located above the lower boundary).”

Comment 10. Extended figure 10: The data on the fertilizer as per the provided link (Line 438) goes to 2019. It is not clear how the maps were generated and why the data is only up to 2013. The colour scales on Figure 10 are also misleading starting at -2000 tonnes per year for aquaculture. It should start at 0, and go much higher to account for the difference in production for China and Brazil, as per the data on the websites provided by the authors.

Reply: This confusion was caused by our previous writing. We have corrected the text in the revision. “2013” was a typo, and the original timeframe was from 2003 to 2017 (as mentioned in the data sources section), which now has been updated to 2019. Fig 10d presents the trend for aquaculture, with both positive and negative trends in different countries. We have revised the figure caption as “**(b-c) Trends of nitrogen and phosphorus from 2003 to 2019 for different countries. (d) Trends of fishery production from 2003 to 2018.**”

Comment 11. Extended data: Figure 9 there is no (f).

Reply: Corrected. Thanks for catching this typo.

Reviewer Reports on the First Revision:

Referees' comments:

Referee #1 (Remarks to the Author):

I read the revised version of the manuscript and was able to verify that all the comments I made were correctly taken into account by modifying the text and by answering my questions. Scientifically, I would say that this is a well-done work that certainly deserves to be published in Nature.

Referee #2 (Remarks to the Author):

All of my comments and suggestions were sufficiently addressed in the revised manuscript. I recommend publication.

Referee #3 (Remarks to the Author):

I believe the authors have done a commendable job on revising the manuscript and addressing my original comments.

Referee #4 (Remarks to the Author):

The authors have answered my queries and I have no further comments.

Author Rebuttals to First Revision:

Referees' comments:

Referee #1 (Remarks to the Author):

I read the revised version of the manuscript and was able to verify that all the comments I made were correctly taken into account by modifying the text and by answering my questions. Scientifically, I would say that this is a well-done work that certainly deserves to be published in Nature.

Referee #2 (Remarks to the Author):

All of my comments and suggestions were sufficiently addressed in the revised manuscript. I recommend publication.

Referee #3 (Remarks to the Author):

I believe the authors have done a commendable job on revising the manuscript and addressing my original comments.

Referee #4 (Remarks to the Author):

The authors have answered my queries and I have no further comments.

Reply: We thank all the referees for their recommendation.